# BE AFFECTIVE, NOT JUST COGNITIVE - TOWARDS IMPARTING PERTINENT EMPATHY IN DIALOGUE AGENTS

## ABSTRACT

Empathetic Response Generation (ERG) has gained significant attention in diverse areas but still faces challenges that hinder its effectiveness. These challenges include 1) the lack of affective empathy in existing works, where they exhibit cognitive empathy (feel *for* user); 2) generate generic responses, where agents address an emotion with monotonous replies; 3) have limited user relatability. To tackle these issues, we propose incorporating affective empathy in models through additional pre-training. We introduce a benchmark dataset and its collection mechanism, that helps curate an 8.5GB dataset, enabling the agent to truly feel *with* user. Using this pre-trained model, our framework EMPATH enhances ERG by reducing generic responses. This is achieved by a novel loss function that involves both conversation history and golden response. EMPATH also enhances user relatability by accounting for multiple emotions and their underlying causes via explainability. Through extensive experimentation, we demonstrate the effectiveness of our dataset on our proposed framework and other existing approaches. Additionally, we depict EMPATH's superior performance in ERG on benchmark datasets across various metrics.

## 1 INTRODUCTION

Empathetic Response Generation (ERG) is the capability of dialogue agents to understand the user's emotions and respond with care and sensitivity, a process known as demonstrating empathy (Davis, 2006), leading to systems where users feel acknowledged (Sinclair et al., 2017; Barker et al., 2023). ERG has gained traction across a variety of fields, including healthcare (Martins et al., 2024), customer service (Leocádio et al., 2024), mental health (Clark & Bailey, 2024), and marketing (Israfilzade & Sadili, 2024). As illustrated in Figure 1, this growing interest in ERG has shed light on a few persistent issues.

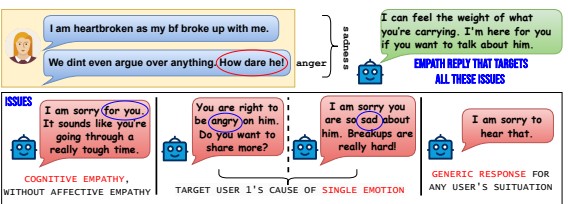

Figure 1: Conversation snippets highlighting various flaws in existing ERG works.

*1.* **Lack of affective empathy**: Existing ERG models primarily focus on recognizing emotions and providing emotionally supportive responses in various contexts, a process called cognitive empathy (Davis, 2006; Jeffrey, 2016). Works like (Sabour et al., 2022; Hsu et al., 2023; Sorin et al., 2024) demonstrate that existing

ERG models (Majumder et al., 2020; Shi et al., 2024; Majumder et al., 2022) and even advanced LLMs like ChatGPT (OpenAI et al., 2024), Gemini (Team et al., 2024), etc exhibit cognitive empathy (Figure 4 in Appendix). Cognitive empathy helps agents recognize users' misery (Figure 1), but responses like "I'm sorry to hear you feel that way" reflect sympathy, not empathy, as renowned psychologist Dr. Brené Brown argues. In her book **Dare To Lead** (Brown, 2018), Brown explains that genuine empathy (also called affective empathy (Davis, 2006; Jeffrey, 2016)) is to feel with people and connect with responses like "I get it" or "I feel with you". In contrast, sympathy feels for them and creates a sense of distance with phrases like "I feel sorry for you" or "I'm sorry you feel that way". Furthermore, several studies (Jeffrey, 2016; Sinclair et al., 2017; Barker et al., 2023) have proven that patients rejected sympathy adversely (Hindu, 2025).

*II*. **Generic responses and user relatability**: Previous works (Majumder et al., 2020; Li et al., 2020; Shi et al., 2024; Pang et al., 2023) that demonstrate cognitive empathy, often produce monotonous, generic responses. In any sad scenario, like in Figure 1, they tend to generate statements like "I am sorry to hear that," which lack genuine empathy and fail to connect with user's emotional needs. To mitigate generic responses, user relatability was enhanced by inculcating the cause of emotion. However, existing Emotion Cause Extraction (ECE) research in ERG primarily focuses on evaluating similarity and fluency against a single golden response and mainly targets a single emotion's cause: either focuses on the entire dialogue's emotion (Li et al., 2021; Gao et al., 2021; Qian et al., 2023a; Liu et al., 2021) (sadness in Figure 1) or the last utterance's emotion (Wang et al., 2021; Gupta & Dandapat, 2023a) (anger in Figure 1). This narrow focus neglects the multifaceted nature of emotions, resulting in responses that favor a single emotion and are detached from the conversation history, leading to inadequate support for the user's emotional state.

To mitigate the aforementioned shortcomings, we propose a jointly modeled solution involving a benchmark affective empathy dataset, its curation mechanism, and a novel framework, **EMPATH**, that comprehensively tackles these challenges. To elaborate, our solution indulges the following: 1) **Integrating affective empathy**: The models can be additionally pre-trained using a specialized, vast affective empathy dataset to avoid the generation of sympathetic responses. Hence, we present a novel approach for curating this dataset that uses ChatGPT (Welivita & Pu, 2024) and paraphrasing techniques to augment a rich 8.5GB synthetic training dataset, a key contribution of this work. 2) **Avoid generic responses**: Using the additionally pre-trained model, generic responses can be avoided, and user reliability can be improved by taking multiple emotions into account. EMPATH prioritizes them based on their intensity and relative occurrence position to focus on the relevant emotions, as not all of them carry equal significance in a dialogue. Utilizing this refined set of emotions, we weigh the importance of their causes via explainability. To further enhance user reliability and avoid generic responses, unlike existing works, we propose a novel attention-based loss function that ensures the generated response relates to the emotion of the golden response and contains relevant information from the historical context of the dialogue, resulting in a more empathetic and non-generic ERG.

Our contributions in this work are: 1) We present a benchmark dataset and a novel dataset curation mechanism to generate large-scale synthetic data, enabling models to exhibit affective and cognitive empathy. 2) Our proposed framework, EMPATH, generates an empathetic response using a T5-model additionally pre-trained on our dataset by utilizing multiple relevant emotions, their causes, and a novel ERG-specific loss function. 3) Extensive experimentation depicts the superiority of inculcating affective empathy using our dataset in existing works and our proposed framework. Furthermore, results also portray the efficacy of EMPATH in ERG on both quantitative and qualitative grounds.

## 2 RELATED WORK

To enhance the ERG task, early efforts primarily focused on improving the emotional understanding of the model (Majumder et al., 2020; Pang et al., 2023) and enhancing the generation strategies of these models (Rashkin et al., 2019; Xie & Pu, 2021) and including external factors, such as common-sense knowledge (Zhong et al., 2021; Chen et al., 2024), pragmatics (Kim et al., 2021), personality traits (Huang et al., 2024;

Li et al., 2024; Cai et al., 2024). The persistence of issues like genericness and lower user-relatability led to the traction of including the causes of the emotions worked best. Existing works target emotions related to the entire conversation (Li et al., 2021; Gao et al., 2021; Qian et al., 2023b) or specific to the most recent utterance (Gupta & Dandapat, 2023a; Wang et al., 2021). However, these existing works in ERG have predominantly emphasized cognitive empathy, thereby producing sympathetic rather than truly empathetic responses. Even studies (Sabour et al., 2022; Hsu et al., 2023) that attempted to incorporate affective empathy. face the limitation of targeting a single emotion, further reducing the relatability of the responses. To this end, we propose EMPATH, designed to generate empathetic, non-generic, and user-relatable responses that embrace cognitive and affective empathy.

## 3 DATASET

In this Section, we curate our dataset[1] by first extracting empathetic sentences from ChatGPT$-4$ and then paraphrasing them up to only two levels (indicating a hierarchy of paraphrasing process in Figure 2) due to our constrained computational resources (more details in Appendix D).

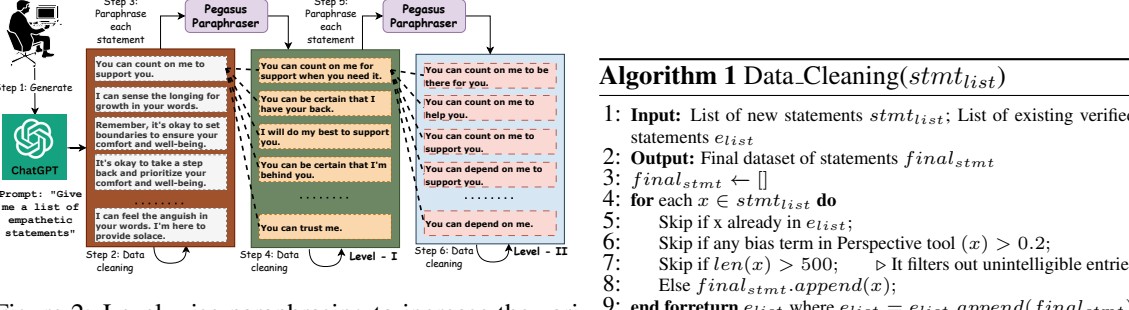

**Algorithm 1** Data_Cleaning($stmt_{list}$)

1: **Input:** List of new statements $stmt_{list}$; List of existing verified statements $e_{list}$
2: **Output:** Final dataset of statements $final_{stmt}$
3: $final_{stmt} \leftarrow []$
4: **for** each $x \in stmt_{list}$ **do**
5:     Skip if x already in $e_{list}$;
6:     Skip if any bias term in Perspective tool $(x) > 0.2$;
7:     Skip if $len(x) > 500$;     ▷ It filters out unintelligible entries
8:     Else $final_{stmt}.append(x)$;
9: **end forreturn** $e_{list}$ where $e_{list} = e_{list}.append(final_{stmt})$

Figure 2: Level-wise paraphrasing to increase the variety of empathetic statements.

### 3.1 DATA COLLECTION

This preliminary step focuses on collecting a large number of empathetic statements, unavailable in existing human-written empathy datasets as they overwhelmingly contain cognitive empathy. So initially, we examined books on empathy by renowned psychologists, but these resources focused on demonstrating empathy rather than providing actual empathetic statements. Consequently, following (Welivita & Pu, 2024), we turned to ChatGPT to generate diverse empathetic expressions. We began by clearly explaining Brown's definition of empathy, ensuring that ChatGPT would focus on generating empathetic rather than sympathetic statements. Following the basics of prompt engineering[2], we then repeatedly issued the query: `Give me a list of empathetic statements`. At each iteration, it consistently produced 250 unique statements, per iteration, before abruptly stopping. This process continued until repetition emerged, i.e., after 500 iterations. We perform a two-fold verification of this dataset. First, we personally review the dataset to ensure it aligns with Brown's concept of empathy and is free from any potential bias or duplicates. Second, to reaffirm that all of ChatGPT's bias (OpenAI, 2024) is removed, we assess each statement using the **Perspective** tool (Jigsaw & Google, 2021), for various bias traits, where lower scores indicate better results. From Table 9 we observe that none of the parameters have scored more than 0.2, indicating a negligible amount of bias. Hence, post data cleaning, we compile a dataset of nearly $111K$ unique, empathetic statements (7.44 MB).

---

[1]https://anonymous.4open.science/r/EMPATH-E6D6/
[2]https://www.promptingguide.ai/introduction/tips

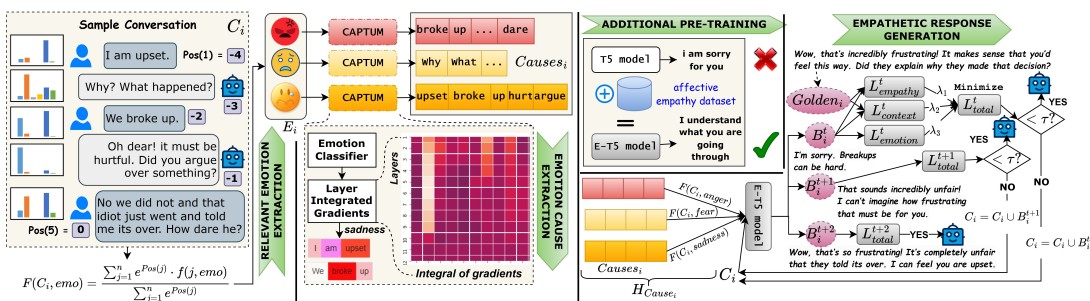

Figure 3: Architecture of EMPATH depicting how a conversation is processed to generate the next response.

## 3.2 DATA PREPARATION

The dataset of $111k$ empathetic statements from ChatGPT was inadequate for the pre-training phase. To enhance our dataset, we propose to expand it through *paraphrasing* each statement using the **Pegasus** (Zhang et al., 2020) model (Table 8 in Appendix D). Each original statement is sent to Pegasus to generate 100 (Appendix D) paraphrased variations, which are stored as the $level_1$ dataset. The Data_Cleaning($level_1$) function in Algorithm 1 results in a dataset of $1.6M$ statements (116 MB) including $level_1$. Using this algorithm, we perform length filtering as extremely long responses generated unintelligible words and de-duplication prevents repeated emotional clichés so the model does not learn templated empathy. Next, we repeat a similar process to obtain the $level_2$ dataset by generating another 100 paraphrased statements for each entry in $level_1$. After bias, duplicates, and unintelligible statements verification using Data_Cleaning($level_2$), we obtain a final dataset of $81.35M$ empathetic statements (8.502 GB) of minimal bias (Appendix D).

## 3.3 DATA VERIFICATION

The final step involves verifying this dataset as automated filters ensure surface quality, but affective empathy requires nuanced emotional resonance that only humans can validate. Given its substantial size, we employ a pool-based active learning (Settles, 2009) based on (Perlitz et al., 2023) to iteratively select the most valuable instances for labeling by our chosen verifiers (Appendix B). We allocate 1% of the dataset to all of the verifiers, with the objective of labeling each sentence as empathetic (1) or not (0). The final label is determined through majority voting. Next, we train a machine learning model on this labeled dataset to perform binary classification on the batch-wise unlabelled data. The human verifiers assess the model's classifications, and we retrain the model iteratively (18 times) until it achieves an acceptable level of performance. This rigorous verification process ensures the integrity of the final dataset, sized 8.5 GB.

## 4 EMPATH

This section presents the EMPATH framework (Figure 3), aimed at enhancing affective empathy in language models. We pre-train a T5-base model on our 8.5GB synthetic dataset, creating the E-T5 model. This model is then used for ERG to analyze causes of multiple emotions in conversations, ensuring the agent focuses on core emotions while minimizing distractions from transient ones. To illustrate, consider a conversation $C_i$ consisting of an $n$-turn dialogue between a speaker $A$ and dialogue agent $B$, structured as $C_i = \{A_i^1, B_i^1, A_i^2, .., A_i^n\}$, where $i$ denotes the dialogue number. Our goal is to generate the dialogue agent's response $B_i^n$ by providing a composite input of the conversation history $[C_i]$, and the weighted (based on relevant emotions $[E_i]$) cause(s) $[Causes_i]$ of each emotion derived using explainability.

## 4.1 Affective Empathy Model

To build E-T5 within the EMPATH framework, we additionally pre-train the T5 model due to its strong performance (Table 4) on diverse datasets and balance of empathy (Appendix G). It is pre-trained for an extra 200k steps to optimize the pre-trained weights from T5 in the context of affective empathetic literature. We set the batch size as 16, the learning rate as $1e-3$, the corruption rate at $5\%$, and a sequence length of 512 tokens for input and output. We utilize the Nvidia K80/T4 GPU of 80 GB RAM (25 days) using Pytorch to finalize these parameters after extensive hyperparameter tuning (Figure 8 in Appendix). Given our limited computing resources, we are starting with the base model, with other variants as future developments.

## 4.2 Relevant Emotion Identification

Accurate emotion detection is crucial for ERG, as the addressed emotion significantly influences the process. Our work emphasizes the importance of considering multiple contextually relevant emotions when crafting non-generic empathetic responses. For instance, in the conversation shown in Figure 3, the utterance $B_i^1$ reflects 'fear' and 'surprise' as prominent emotions at that moment. However, 'surprise' is not prevalent throughout the entire conversation. Building on that intuition, we negate the influence of less prevailing emotions observed during the conversation based on the position of each utterance using Equation 1. Following (Lin et al., 2019), we find the probability distribution for each utterance across six emotions (anger, fear, love, joy, surprise, and sadness, based on (Gupta et al., 2025)). Using Equation 1, we obtain the final probability of each emotion and pick those as relevant emotions with a value $> 0.75$ (threshold)[3].

$$F(C_i, emo) = \frac{\sum_{j=1}^{n} e^{Pos(j)} \cdot f(j, emo)}{\sum_{j=1}^{n} e^{Pos(j)}} \tag{1}$$

where $F(C_i, emo)$ represents the final value for a particular emotion $emo$ in the conversation $C_i$, $Pos$ represents the position, and $j$ represents the utterance of the $i^{th}$ conversation. In our approach, we label the most recent utterance as position 0, with prior utterances labeled as $-1, -2$, etc., based on their relative distance from it. The expression, $F(C_i, emo)$ is calculated using the average weighted sum of the value in the probability distribution for $emo$ in utterance $j$, represented by $f(j, emo)$, and the exponentiated position of that utterance $e^{Pos(j)}$. This results in the set of relevant emotions, $E_i = \{E_i^1, E_i^2, ..., E_i^k\}$, where $k$ is the number of emotions greater than the threshold.

## 4.3 Emotion Cause Extraction

The existing approaches on ERG utilizing ECE treat each utterance as a collection of clauses, aiming to identify the cause clause. This approach has a notable limitation, as the length of a clause is highly contingent on the delimiters employed; it can result in disproportionately long clauses. When these lengthy clauses are predicted as cause clauses, they inadvertently emphasize unimportant words, thereby obscuring the actual emotional causality. To mitigate this issue, we propose using **explainability** to extract concise causes of multiple emotions. Unlike clause-based methods, explainability provides evidence for model outcomes without requiring clause extraction. As shown in Figure 3, the entire conversation is fed into an explainability module for emotion classification. We analyze which parts of the conversation contribute to detecting relevant emotions. For classification, we fine-tune a BERT model on the Emotions dataset for NLP[4]. To evaluate content contribution, we use *Layer Integrated Gradients* from the **Captum** library (Kokhlikyan et al., 2020). The words with a score $> 0.7$ (darkly highlighted)[3] are identified as causes and prioritized for generating empathetic responses. So $Causes_i = \{Cause_i^{1,1}, Cause_i^{2,1}, ..., Cause_i^{1,2}, ...\}$, where $Cause_i^{x,k}$ represents the cause word $x$ of the $k^{th}$ emotion. Our goal is to enhance user relatability by incorporating multiple relevant emotions. Therefore, we prioritize the causes of the most pertinent emotions, weighting them according

---

[3]We reach this threshold by thorough empirical analysis.

[4]https://www.kaggle.com/datasets/praveengovi/emotions-dataset-for-nlp

to the importance of their corresponding emotions. So for a particular cause word $x$'s embedding:

$$W_{Cause_i^{x,k}} = w(E_i^k) \times \text{Embed}(Cause_i^{x,k}) \text{ where } w(E_i^k) = F(C_i, k) \tag{2}$$

where $W_{Cause_i^{x,k}}$ would be its weight and $w(E_i^k)$ depicts the weight to be given to each cause of emotion $k$ from $E_i$. To generate an initial empathetic response $B_i^{(t)}$, we provide E-T5 model with the context embedding $H_{context}$ as a concatenated input of conversation history $C_i$ and all the weighted causes, $H_{Cause_i}$.

$$H_{context} = \text{Embed}(C_i) \oplus H_{Cause_i} \text{ where } H_{Cause_i} = \bigoplus_{emo=1}^{k} \bigoplus_{b=1}^{x} W_{Cause_i^{b,emo}} \tag{3}$$

$$B_i^{(t)} = \text{E-T5}(H_{context}) \tag{4}$$

## 4.4 EMOTION-AWARE CONTEXTUAL LOSS

For the initial response, $B_i^{(t)}$; however, we cannot ascertain whether it is empathetic, relevant, and non-generic or not. To evaluate this, we calculate a loss function, $L_{total}^t$ that considers three aspects: **Emotion Alignment Loss** ($L_{emotion}^t$): To prevent misaligned emotions, we align the generated response with the emotional tone of the golden response by calculating the KL Divergence between their emotion distributions (obtained using the method from Section 4.2).

$$L_{emotion}^t = \sum_{emo=1}^{k} f(Golden_i, emo) \log \frac{f(Golden_i, emo)}{f(B_i^{(t)}, emo)} \tag{5}$$

**Contextual Relevance Loss** ($L_{context}^t$): To ensure the generated response is not generic, we penalize responses that don't align with the conversation history. This forces responses to reference past dialogue, by measuring how dissimilar they are with the history using the cosine similarity.

$$L_{context}^t = 1 - \frac{\text{Embed}(B_i^{(t)}) \cdot \text{Embed}(C_i)}{\overline{\text{Embed}(B_i^{(t)})}|| \ ||\text{Embed}(C_i)||} \tag{6}$$

**Empathy Validation Loss** ($L_{empathy}^t$): To prevent low-empathy generic responses, we utilize the classifier from Section 3.3 to compute the Mean Squared Error between confidence scores of empathy for the golden and generated responses.

$$L_{empathy}^t = (\text{Confidence}(Golden_i) - \text{Confidence}(B_i^t))^2 \tag{7}$$

$$L_{total}^t = \lambda_1 L_{emotion}^t + \lambda_2 L_{context}^t + \lambda_3 L_{empathy}^t \tag{8}$$

The equation for the total loss function ($L_{total}^t$) incorporates these losses, weighted by hyperparameters $\lambda_1, \lambda_2, \lambda_3$. These hyperparameters control the relative importance of each component in the overall loss calculation. During optimization, at each iteration $t$, the hyperparameters are updated jointly with the model parameters using the Adam optimizer, based on the gradients of the total loss with respect to each $\lambda$. This iterative process allows the model to fine-tune the balance between emotion, context, and empathy, ensuring that each aspect contributes effectively to the overall performance.

## 4.5 EMPATHETIC RESPONSE GENERATION

The total loss, is computed by aggregating the individual loss components at each iteration. If $L_{total}^t < \tau$ then $B_i^n = B_i^{(t)}$, else response is regenerated. To do so, we first update the context set by adding the generated response to it, $C_i \leftarrow C_i \cup B_i^{(t)}$. Next, the updated context embedding, $H_{context}^{t+1}$, is computed by concatenating the embeddings of the updated context $C_i$ and current cause embedding $H_{Cause_i}$, to generate the next response, $B_i^{(t+1)}$.

$$H_{context}^{t+1} = \text{Embed}(C_i) \oplus H_{cause_i} \tag{9}$$

$$B_i^{(t+1)} = \text{E-T5}(H_{context}^{t+1}) \tag{10}$$

If the regenerated response still fails to reduce $L_{total}$ below the threshold, the process repeats till,

$$L_{total}^t < \tau \text{ or } L_{total}^t \approx L_{total}^{t+1} \text{ for 10 iterations} \tag{11}$$

## 5 RESULTS AND DISCUSSION

In this Section, we wish to empirically answer the following research questions:

**RQ1**: How effective is synthetic data in integrating affective empathy with cognitive empathy?
**RQ2**: Is explainability a better alternative than clause-based approaches for ECE?
**RQ3**: How effectively does EMPATH address the issue of generic responses in ERG?
**RQ4**: How does incorporating multiple emotions in dialogue agents affect ERG?

### 5.1 BASELINES AND SETUP

In this work, we build two versions of E-T5: **Medium**, trained on $level_1$ data and **Large**, trained on $level_2$ data, and compare our EMPATH framework using both variants with SOTA existing generative models: GPT2 (Radford et al., 2019), DialoGPT (Zhang et al., 2019), GPT-Neo (Gao et al., 2020) (7*B*), T5-base (Raffel et al., 2020); Zero-shot prompted LLMs excluding ChatGPT (as it's used for synthetic data generation and evaluation) like Llama-2 (Touvron et al., 2023), Llama-3 (8B), Claude, Gemini; all SOTA ERG models that consider emotion causes: EMMA (Li et al., 2021), SEEC (Gupta & Dandapat, 2023a), ECTG (Qian et al., 2023b), GATE (Gao et al., 2021), GREC(Wang et al., 2021). We split the English datasets into $8 : 1 : 1$ (train, valid, test) ratio for a fair comparison using these hyperparameters: epochs= $10$, batch size= $16$, the Adam optimizer, learning rate= $1e - 3$, $top\_k = 50$, $top\_p = 0.95$, temperature= $0.85$, $max\_tokens = 250$, context length= default per API and a uniform prompt across LLMs: "Generate an empathetic response to the given conversation") to ensure comparability (more details in Appendix G).

### 5.2 RQ1: SYNTHETIC DATA ANALYSIS

We present the quantitative comparison (single run) among the baselines and E-T5 on benchmark dialogue datasets using standard metrics (Appendix C), and Table 4 depicts the enhancement in the type of response generated after incorporating affective empathy in our proposed E-T5 model. Additionally, raw pre-trained models are compared over a set of 100 randomly chosen conversations from both empathetic real-world dialogue datasets. The increase in the Distinct-n values depicts how effectively E-T5 utilizes the broad vocabulary it is trained on. Comparing results by three annotators (Appendix B) and LLM-as-judge (ChatGPT) reveals a significantly higher win % across datasets and in its raw pre-trained version. Furthermore, the performance of the existing cause-based ERG models and other LLMs like GPT-2 and DialoGPT (Appendix G) also improves by including affective empathy in Table 1.

Table 1: Improvement in models by introducing affective empathy. *Note: ∗ indicates replacing their generative model with E-T5 and # indicates additionally pre-trained on our synthetic dataset.*

**CHASE** (Gupta & Dandapat, 2023a) (test size = 500 conversations)

| Model | PPL(↓) | | Distinct(↑) | | | | | | SES(↑) | | BLEU(↑) | | AL(↑) | | ROUGE F1-Score(↑) | | | | | | METEOR(↑) | |
|---|---|---|---|---|---|---|---|---|---|---|---|---|---|---|---|---|---|---|---|---|---|---|
| | w/o | w | 1 (w/o) | 1 (w) | 2 (w/o) | 2 (w) | 3 (w/o) | 3 (w) | w/o | w | w/o | w | w/o | w | 1 (w/o) | 1 (w) | 2 (w/o) | 2 (w) | L (w/o) | L (w) | w/o | w |
| EMMA∗ | 19.778 | 15.660 | 0.552 | 0.689 | 0.895 | 0.983 | 0.938 | 0.943 | 0.737 | 0.751 | 0.142 | 0.156 | 7 | 8 | 0.146 | 0.225 | 0.111 | 0.195 | 0.200 | 0.214 | 0.130 | 0.238 |
| ECTG∗ | 32.769 | 23.336 | 0.567 | 0.704 | 0.945 | 0.973 | 0.932 | 0.937 | 0.398 | 0.412 | 0.154 | 0.168 | 7 | 7 | 0.219 | 0.298 | 0.083 | 0.167 | 0.250 | 0.264 | 0.098 | 0.206 |
| GATES∗ | 35.363 | 25.930 | 0.280 | 0.417 | 0.754 | 0.842 | 0.894 | 0.899 | 0.543 | 0.557 | 0.133 | 0.147 | 15 | 13 | 0.235 | 0.314 | 0.081 | 0.165 | 0.196 | 0.210 | 0.145 | 0.253 |
| GREC∗ | 27.989 | 18.556 | 0.400 | 0.537 | 0.725 | 0.813 | 0.775 | 0.780 | 0.722 | 0.736 | 0.180 | 0.194 | 10 | 11 | 0.416 | 0.495 | 0.272 | 0.356 | 0.400 | 0.430 | 0.384 | 0.456 |
| SEEC∗ | 16.004 | 10.091 | 0.542 | 0.679 | 0.885 | 0.973 | 0.913 | 0.947 | 0.855 | 0.908 | 0.325 | 0.339 | 16 | 17 | 0.421 | 0.479 | 0.152 | 0.227 | 0.210 | 0.244 | 0.355 | 0.463 |
| GPT-2 (small)# | 18.251 | 15.778 | 0.698 | 0.708 | 0.926 | 0.933 | 0.896 | 0.924 | 0.67 | 0.782 | 0.250 | 0.349 | 4 | 13 | 0.261 | 0.349 | 0.142 | 0.349 | 0.25 | 0.349 | 0.146 | 0.156 |
| GPT-2 (medium)# | 18.539 | 15.247 | 0.797 | 0.799 | 0.928 | 0.934 | 0.856 | 0.916 | 0.685 | 0.793 | 0.333 | 0.333 | 4 | 17 | 0.106 | 0.311 | 0.008 | 0.347 | 0.117 | 0.333 | 0.043 | 0.142 |
| DialoGPT (small)# | 18.775 | 14.975 | 0.431 | 0.646 | 0.768 | 0.821 | 0.847 | 0.911 | 0.555 | 0.689 | 0.095 | 0.115 | 14 | 16 | 0.129 | 0.187 | 0.105 | 0.187 | 0.103 | 0.187 | 0.095 | 0.182 |
| DialoGPT (medium)# | 16.334 | 14.555 | 0.352 | 0.583 | 0.792 | 0.803 | 0.929 | 0.931 | 0.239 | 0.444 | 0.058 | 0.095 | 13 | 16 | 0.280 | 0.261 | 0.114 | 0.142 | 0.212 | 0.250 | 0.260 | 0.339 |

**EmpatheticDialogue** (Rashkin et al., 2019) (test size = 2.4k conversations)

| Model | PPL(↓) | | Distinct(↑) | | | | | | SES(↑) | | BLEU(↑) | | AL(↑) | | ROUGE F1-Score(↑) | | | | | | METEOR(↑) | |
|---|---|---|---|---|---|---|---|---|---|---|---|---|---|---|---|---|---|---|---|---|---|---|
| | w/o | w | 1 (w/o) | 1 (w) | 2 (w/o) | 2 (w) | 3 (w/o) | 3 (w) | w/o | w | w/o | w | w/o | w | 1 (w/o) | 1 (w) | 2 (w/o) | 2 (w) | L (w/o) | L (w) | w/o | w |
| EMMA∗ | 29.666 | 21.531 | 0.309 | 0.488 | 0.785 | 0.989 | 0.880 | 0.889 | 0.524 | 0.536 | 0.090 | 0.196 | 10 | 11 | 0.166 | 0.195 | 0.090 | 0.197 | 0.166 | 0.265 | 0.096 | 0.184 |
| ECTG∗ | 35.801 | 27.666 | 0.333 | 0.512 | 0.789 | 0.925 | 0.912 | 0.921 | 0.414 | 0.426 | 0.076 | 0.182 | 13 | 13 | 0.222 | 0.251 | 0.076 | 0.183 | 0.222 | 0.321 | 0.125 | 0.213 |
| GATES∗ | 27.536 | 19.401 | 0.516 | 0.695 | 0.903 | 0.932 | 0.935 | 0.944 | 0.168 | 0.180 | 2.760 | 2.866 | 7 | 6 | 0.121 | 0.150 | 0.152 | 0.259 | 0.060 | 0.159 | 0.061 | 0.149 |
| GREC∗ | 24.180 | 16.045 | 0.410 | 0.589 | 0.770 | 0.974 | 0.854 | 0.863 | 0.721 | 0.733 | 2.130 | 2.236 | 10 | 10 | 0.416 | 0.445 | 0.200 | 0.307 | 0.416 | 0.515 | 0.340 | 0.428 |
| SEEC∗ | 10.041 | 9.906 | 0.633 | 0.812 | 0.936 | 0.911 | 0.953 | 0.962 | 0.802 | 0.814 | 2.066 | 2.172 | 11 | 11 | 0.422 | 0.459 | 0.286 | 0.373 | 0.351 | 0.399 | 0.338 | 0.426 |
| GPT-2 (small)# | 17.342 | 14.365 | 0.705 | 0.741 | 0.941 | 0.949 | 0.882 | 0.892 | 0.065 | 0.413 | 0.180 | 0.321 | 6 | 15 | 0.066 | 0.146 | 0.171 | 0.111 | 0.066 | 0.2 | 0.02 | 0.151 |
| GPT-2 (medium)# | 22.483 | 18.952 | 0.328 | 0.597 | 0.863 | 0.902 | 0.958 | 0.960 | 0.696 | 0.753 | 0.142 | 1.347 | 11 | 16 | 0.076 | 0.129 | 0.166 | 0.205 | 0.076 | 0.103 | 0.094 | 0.172 |
| DialoGPT (small)# | 15.037 | 12.983 | 0.440 | 0.567 | 0.920 | 0.935 | 0.960 | 0.961 | 0.320 | 0.549 | 0.209 | 0.333 | 11 | 17 | 0.162 | 0.166 | 0.090 | 0.166 | 0.162 | 0.166 | 0.080 | 0.157 |
| DialoGPT (medium)# | 16.113 | 12.746 | 0.416 | 0.582 | 0.875 | 0.928 | 0.937 | 0.955 | 0.551 | 0.633 | 0.245 | 0.329 | 10 | 18 | 0.232 | 0.286 | 0.098 | 0.280 | 0.143 | 0.286 | 0.031 | 0.120 |

Table 2: Efficiency of Explainability in ECE task.

| Model | Automatic Assessment | | | | Human | LLM-as-judge |
|---|---|---|---|---|---|---|
| | Precision | Recall | F1-Score | Conciseness | (Average) | (ChatGPT) |
| E-2D (BERT) (Ding et al., 2020a) | 0.393 | 0.639 | 0.486 | 0.556 | 0.390 | 0.351 |
| E-MLL (BERT) (Ding et al., 2020b) | 0.255 | 0.650 | 0.366 | 0.582 | 0.402 | 0.394 |
| RTHN (Xia et al., 2019) | 0.428 | 0.598 | 0.498 | 0.366 | 0.319 | 0.258 |
| SCAECE (Gupta & Dandapat, 2023b) | **0.552** | 0.648 | 0.596 | 0.626 | 0.446 | 0.388 |
| KJ-IECE (Wu et al., 2023) | 0.367 | 0.563 | 0.444 | 0.645 | 0.465 | 0.381 |
| Explainability | 0.526 | **0.730** | **0.611** | **0.851** | **0.744** | **0.721** |

Table 3: Performance on EmpatheticDialogue dataset without (w/o) each component.

| EMPATH | PPL | Distinct | | | SES | BLEU | AL | ROUGE F1-Score | | | METEOR |
|---|---|---|---|---|---|---|---|---|---|---|---|
| (large) | | 1 | 2 | 3 | | | | 1 | 2 | L | |
| w/o emotions | 18.567 | 0.648 | 0.845 | 0.771 | 0.359 | 0.041 | 8 | 0.250 | 0.281 | 0.310 | 0.250 |
| w/o causes | 17.901 | 0.555 | 0.602 | 0.631 | 0.291 | 0.031 | 8 | 0.185 | 0 | 0.185 | 0.155 |
| w/o $L_{emotion}$ | 12.579 | 0.677 | 0.745 | 0.777 | 0.456 | 1.030 | 7 | 0.159 | 0.163 | 0.197 | 0.212 |
| w/o $L_{context}$ | 14.476 | 0.687 | 0.739 | 0.755 | 0.368 | 1.076 | 7 | 0.129 | 0.147 | 0.183 | 0.301 |
| w/o $L_{empathy}$ | 15.666 | 0.653 | 0.697 | 0.726 | 0.309 | 1.008 | 9 | 0.129 | 0.027 | 0.151 | 0.244 |
| w/o $L_{total}$ | 17.912 | 0.604 | 0.666 | 0.711 | 0.297 | 0.961 | 17 | 0.117 | 0 | 0.054 | 0.116 |
| w/o E-T5 | 25.580 | 0.347 | 0.477 | 0.523 | 0.087 | 0.400 | 7 | 0.018 | 0.016 | 0.141 | 0.062 |

Table 4: Comparison results of generative models in their pre-trained and fine-tuned version.

| Model | | DailyDialog (Li et al., 2017) (test size = 1k conversations) | | | | | | | | | | Persona-Chat (Zhang et al., 2018) (test size = 700 conversations) | | | | | | | | | | |
|---|---|---|---|---|---|---|---|---|---|---|---|---|---|---|---|---|---|---|---|---|---|---|
| | | PPL(↓) | Distinct(↑) | | | AL(↑) | Human A/B Testing | | | LLM-as-judge | | | PPL(↓) | Distinct(↑) | | | AL(↑) | Human A/B Testing | | | LLM-as-judge | | |
| | | | 1 | 2 | 3 | | Win | Loss | Tie | Win | Loss | Tie | | 1 | 2 | 3 | | Win | Loss | Tie | Win | Loss | Tie |
| GPT-2 | small | 16.155 | 0.647 | 0.882 | 0.882 | 4 | 75.2 | 20.0 | 4.8 | 70.1 | 18.3 | 11.6 | 19.847 | 0.423 | 0.884 | 0.961 | 11 | 77.8 | 18.2 | 4.0 | 71.9 | 20.1 | 8.0 |
| | medium | 17.267 | 0.533 | 0.900 | 0.933 | 6 | 65.1 | 15.3 | 19.6 | 68.2 | 15.0 | 16.8 | 19.889 | 0.411 | 0.666 | 0.333 | 9 | 66.9 | 15.3 | 17.8 | 70.8 | 15.0 | 14.2 |
| | large | 17.899 | 0.777 | 0.947 | 0.888 | 8 | 78.1 | 18.2 | 3.7 | 66.8 | 13.2 | 20.0 | 20.001 | 0.400 | 0.600 | 0.640 | 10 | 80.7 | 16.4 | 2.9 | 68.6 | 15.0 | 16.4 |
| DialoGPT | small | 14.754 | 0.298 | 0.807 | 0.924 | 19 | 72.9 | 23.1 | 4.0 | 71.4 | 11.0 | 17.6 | 17.198 | 0.377 | 0.751 | 0.698 | 11 | 74.7 | 23.1 | 2.2 | 74.0 | 11.0 | 15.0 |
| | medium | 16.652 | 0.684 | 0.888 | 0.794 | 10 | 62.8 | 18.4 | 18.8 | 65.3 | 14.5 | 20.2 | 17.567 | 0.395 | 0.609 | 0.653 | 11 | 65.4 | 16.6 | 18.0 | 67.1 | 16.3 | 16.6 |
| | large | 16.888 | 0.588 | 0.823 | 0.782 | 9 | 75.8 | 21.3 | 2.9 | 75.3 | 18.4 | 6.3 | 17.909 | 0.366 | 0.466 | 0.308 | 13 | 77.6 | 21.3 | 1.1 | 77.9 | 18.4 | 3.7 |
| GPT-Neo | | 56.874 | 0.375 | 0.588 | 0.672 | 12 | 80.4 | 6.6 | 13.0 | 75.4 | 13.0 | 11.6 | 49.263 | 0.280 | 0.631 | 0.714 | 20 | 83.0 | 4.8 | 12.2 | 77.2 | 14.8 | 8.0 |
| T5-base | | 17.065 | 0.812 | 0.923 | 0.933 | 7 | 55.6 | 13.7 | 30.7 | 60.4 | 25.5 | 14.1 | 18.001 | 0.562 | 0.822 | 0.909 | 8 | 57.4 | 13.7 | 28.9 | 63.0 | 25.5 | 11.5 |
| Llama-2 | 7B | 32.947 | 0.389 | 0.501 | 0.665 | 125 | 43.3 | 20.0 | 36.7 | 40.7 | 15.5 | 43.8 | 25.198 | 0.376 | 0.344 | 0.671 | 95 | 35.9 | 28.2 | 35.9 | 42.5 | 17.3 | 40.2 |
| Llama-3 | 8B | 36.044 | 0.400 | 0.510 | 0.650 | 151 | 44.7 | 23.5 | 31.8 | 38.8 | 12.9 | 48.3 | 27.643 | 0.415 | 0.439 | 0.724 | 90 | 36.5 | 23.5 | 40.0 | 41.4 | 12.9 | 45.7 |
| Claude | 4.5 | 38.127 | 0.207 | 0.661 | 0.838 | 126 | 42.3 | 31.0 | 26.7 | 46.8 | 14.7 | 38.5 | 25.198 | 0.163 | 0.644 | 0.841 | 86 | 44.4 | 15.2 | 40.4 | 47.5 | 12.3 | 40.2 |
| Gemini | 2.5 | 41.297 | 0.059 | 0.343 | 0.698 | 110 | 43.2 | 39.7 | 17.1 | 41.5 | 11.0 | 47.5 | 47.125 | 0.026 | 0.189 | 0.486 | 216 | 46.6 | 13.5 | 39.9 | 44.3 | 15.0 | 40.7 |
| E-T5 | medium | 12.845 | 0.813 | 0.952 | **0.947** | 13 | - | - | - | - | - | - | 10.480 | 0.545 | 0.878 | **0.911** | 6 | - | - | - | - | - | - |
| | large | **11.902** | **0.846** | **0.984** | 0.916 | 9 | - | - | - | - | - | - | **10.111** | **0.717** | **0.918** | 0.886 | 9 | - | - | - | - | - | - |

| Model | | Blended Skill Talk (Smith et al., 2020) (test size = 700 conversations) | | | | | | | | | | Pre-Trained | | | | | | | | | | |
|---|---|---|---|---|---|---|---|---|---|---|---|---|---|---|---|---|---|---|---|---|---|---|
| | | PPL(↓) | Distinct(↑) | | | AL(↑) | Human A/B Testing | | | LLM-as-judge | | | PPL(↓) | Distinct(↑) | | | AL(↑) | Human A/B Testing | | | LLM-as-judge | | |
| | | | 1 | 2 | 3 | | Win | Loss | Tie | Win | Loss | Tie | | 1 | 2 | 3 | | Win | Loss | Tie | Win | Loss | Tie |
| GPT-2 | small | 25.199 | 0.500 | 0.631 | 0.684 | 7 | 77.0 | 16.8 | 6.2 | 68.3 | 20.6 | 11.1 | 26.854 | 0.411 | 0.723 | 0.821 | 10 | 81.1 | 18.5 | 0.4 | 72.6 | 16.8 | 10.6 |
| | medium | 26.190 | 0.538 | 0.697 | 0.717 | 9 | 66.9 | 13.0 | 20.1 | 66.4 | 17.3 | 16.3 | 28.107 | 0.565 | 0.856 | 0.800 | 4 | 69.4 | 13.8 | 16.8 | 74.1 | 11.7 | 14.2 |
| | large | 26.555 | 0.563 | 0.684 | 0.736 | 7 | 79.9 | 15.0 | 5.1 | 65.0 | 15.5 | 19.5 | 24.728 | 0.500 | 0.857 | 0.928 | 7 | 84.0 | 14.9 | 1.1 | 69.3 | 11.7 | 19.0 |
| DialoGPT | small | 22.841 | 0.600 | 0.860 | 0.820 | 5 | 74.7 | 20.8 | 4.5 | 69.6 | 13.3 | 17.1 | 32.093 | 0.409 | 0.409 | 0.518 | 3 | 77.2 | 21.6 | 1.2 | 77.3 | 7.7 | 15.0 |
| | medium | 22.943 | 0.619 | 0.857 | 0.750 | 6 | 64.6 | 15.2 | 20.2 | 63.5 | 16.8 | 19.7 | 34.321 | 0.432 | 0.862 | 0.891 | 6 | 68.7 | 15.1 | 16.2 | 67.8 | 13.0 | 19.2 |
| | large | 23.167 | 0.666 | 0.871 | 0.800 | 6 | 77.6 | 19.0 | 3.4 | 73.5 | 20.7 | 5.8 | 37.517 | 0.514 | 0.742 | 0.742 | 8 | 80.1 | 19.8 | 0.1 | 81.2 | 15.1 | 3.7 |
| GPT-Neo | | 39.463 | 0.260 | 0.702 | 0.737 | 18 | 82.2 | 3.4 | 14.4 | 73.6 | 15.3 | 11.1 | 40.176 | 0.387 | 0.836 | 0.918 | 11 | 86.3 | 3.3 | 10.4 | 77.9 | 11.5 | 10.6 |
| T5-base | | 13.989 | 0.584 | 0.848 | 0.899 | 8 | 57.4 | 11.4 | 31.2 | 58.6 | 27.8 | 13.6 | 16.267 | 0.600 | 0.800 | 0.900 | 8 | 59.9 | 12.2 | 27.9 | 66.3 | 22.2 | 11.5 |
| Llama-2 | 7B | 40.695 | 0.402 | 0.657 | 0.659 | 156 | 35.1 | 26.8 | 38.1 | 38.9 | 17.8 | 43.3 | 42.189 | 0.053 | 0.307 | 0.634 | 145 | 39.2 | 26.7 | 34.1 | 43.2 | 14.0 | 42.8 |
| Llama-3 | 8B | 44.444 | 0.384 | 0.581 | 0.575 | 170 | 36.5 | 21.2 | 42.3 | 37.0 | 15.2 | 47.8 | 42.091 | 0.047 | 0.291 | 0.578 | 177 | 39.0 | 22.0 | 39.0 | 44.7 | 9.6 | 45.7 |
| Claude | 4.5 | 44.202 | 0.056 | 0.364 | 0.699 | 102 | 31.5 | 10.4 | 58.1 | 38.7 | 17.6 | 43.7 | 43.092 | 0.059 | 0.364 | 0.717 | 94 | 49.2 | 16.7 | 34.1 | 43.6 | 13.7 | 42.7 |
| Gemini | 2.5 | 47.440 | 0.023 | 0.160 | 0.432 | 236 | 46.5 | 11.2 | 42.3 | 46.0 | 10.2 | 43.8 | 48.672 | 0.020 | 0.160 | 0.460 | 375 | 49.0 | 12.0 | 39.0 | 45.7 | 10.6 | 43.7 |
| E-T5 | medium | 11.863 | 0.616 | 0.743 | 0.706 | 9 | - | - | - | - | - | - | 16.968 | 0.750 | 0.863 | **0.931** | 8 | - | - | - | - | - | - |
| | large | **10.044** | **0.743** | **0.892** | **0.928** | 6 | - | - | - | - | - | - | **14.650** | **0.786** | **0.875** | 0.870 | 7 | - | - | - | - | - | - |

## 5.3 RQ2: EXPLAINABILITY FOR ECE

This study evaluates the use of Explainability for the ECE task and compares it with existing SOTA models using the RECCON dataset (Poria et al., 2021). We assess performance using Precision, Recall, F1-score, and a new metric, Conciseness, which measures the ratio of meaningful words to total words, excluding minimal content words (represented by PRP$, VB{P/D}, DT, TO, IN tags). For example, in "I am very happy. I got my promotion today", SOTA models predict the cause clause as "I got my promotion today", while Explainability predicts "promotion". So $Conciseness_{SOTA} = \frac{3}{5} \approx 0.67$ whereas $Conciseness_{ours} : \frac{1}{1} = 1$. We also solicit conciseness from both human annotators and LLM-as-judge. Results in Table 2 suggest that by focusing on key feature contributions, explainability provides a more intuitive understanding of ECE.

## 5.4 RQ3: IMPROVEMENT IN GENERIC RESPONSES

We analyze human ratings for each model on the ERG task using 100 randomly selected conversations from Section 5.2. Verifiers of Section 3.3 assess models on Empathy, Appropriateness, Contextualization, and Correctness (Definitions in Appendix B). Results in Table 6 show that incorporating multiple emotions reduces response genericness and enhances empathy. We also perform A/B testing to compare our model with variants, noting wins, losses, and ties. Our model significantly outperforms others in generating affective responses thereby reducing generic responses and demonstrating strong human appreciation with Fleiss' Kappa ($\kappa$) values indicating substantial to almost perfect agreement between raters.

Table 5: Quantitative analysis against baselines across various metrics.

| | Model | | CHASE (test size = 500 conversations) | | | | | | | | | | | EmpatheticDialogue (test size = 2.4k conversations) | | | | | | | | | | |
|---|---|---|---|---|---|---|---|---|---|---|---|---|---|---|---|---|---|---|---|---|---|---|---|---|
| | | | PPL(↓) | Distinct(↑) 1 | 2 | 3 | SES(↑) | BLEU(↑) | AL(↑) | ROUGE F1-Score(↑) 1 | 2 | L | METEOR(↑) | PPL(↓) | Distinct(↑) 1 | 2 | 3 | SES(↑) | BLEU(↑) | AL(↑) | ROUGE F1-Score(↑) 1 | 2 | L | METEOR(↑) |
| No Emotion | GPT-2 | small | 18.251 | 0.698 | 0.926 | 0.896 | 0.670 | 0.250 | 4 | 0.261 | 0.142 | 0.250 | 0.146 | 17.342 | 0.705 | 0.941 | 0.882 | 0.065 | 0.180 | 6 | 0.066 | 0.171 | 0.066 | 0.020 |
| | | medium | 18.539 | 0.797 | 0.928 | 0.856 | 0.685 | 0.333 | 4 | 0.106 | 0.008 | 0.117 | 0.043 | 22.483 | 0.328 | 0.863 | 0.958 | 0.696 | 0.142 | 11 | 0.076 | 0.166 | 0.076 | 0.094 |
| | | large | 19.240 | 0.350 | 0.842 | 0.929 | 0.461 | 0.115 | 7 | 0.090 | 0.125 | 0.090 | 0.031 | 24.163 | 0.441 | 0.794 | 0.852 | 0.631 | 0.172 | 7 | 0.090 | 0.160 | 0.090 | 0.065 |
| | DialoGPT | small | 18.775 | 0.431 | 0.768 | 0.847 | 0.555 | 0.095 | 14 | 0.129 | 0.105 | 0.103 | 0.095 | 15.037 | 0.440 | 0.920 | 0.960 | 0.320 | 0.209 | 11 | 0.162 | 0.090 | 0.162 | 0.080 |
| | | medium | 16.334 | 0.352 | 0.792 | 0.929 | 0.239 | 0.058 | 13 | 0.280 | 0.114 | 0.212 | 0.260 | 16.113 | 0.416 | 0.875 | 0.937 | 0.551 | 0.245 | 10 | 0.232 | 0.098 | 0.143 | 0.031 |
| | | large | 16.515 | 0.352 | 0.775 | 0.910 | 0.365 | 0.208 | 15 | 0.090 | 0.180 | 0.111 | 0.182 | 23.081 | 0.722 | 0.944 | 0.888 | 0.524 | 0.333 | 6 | 0.105 | 0.086 | 0.105 | 0.066 |
| | GPT-Neo | | 18.111 | 0.295 | 0.752 | 0.919 | 0.496 | 0.153 | 20 | 0.187 | 0 | 0.160 | 0.176 | 25.661 | 0.333 | 0.743 | 0.769 | 0.244 | 0.329 | 12 | 0.263 | 0.083 | 0.210 | 0.120 |
| | T5-base | | 16.085 | 0.596 | 0.887 | 0.931 | 0.798 | 0.277 | 11 | 0.198 | 0 | 0.294 | 0.212 | 11.791 | 0.263 | 0.700 | 0.881 | 0.305 | 0 | 18 | 0.166 | 0 | 0.124 | 0.077 |
| | E-T5 | medium | 14.001 | 0.782 | 0.941 | 0.941 | 0.725 | 0.325 | 8 | 0.380 | 0.266 | 0.380 | 0.370 | 10.244 | 0.781 | 0.941 | 0.952 | 0.785 | 2.166 | 6 | 0.444 | 0.366 | 0.444 | 0.311 |
| | | large | 12.459 | 0.750 | 0.905 | 0.862 | 0.892 | 0.244 | 7 | 0.396 | 0.252 | 0.396 | 0.399 | 11.982 | 0.773 | 0.935 | 0.925 | 0.813 | 3.151 | 7 | 0.413 | 0.320 | 0.413 | 0.322 |
| 1 Emotion | SOTA EMMA | | 19.778 | 0.552 | 0.895 | 0.938 | 0.737 | 0.142 | 7 | 0.146 | 0.111 | 0.200 | 0.130 | 29.666 | 0.309 | 0.785 | 0.880 | 0.524 | 0.090 | 10 | 0.166 | 0.090 | 0.166 | 0.096 |
| | ECTG | | 32.769 | 0.567 | 0.945 | 0.932 | 0.398 | 0.154 | 7 | 0.219 | 0.083 | 0.250 | 0.098 | 35.801 | 0.333 | 0.789 | 0.912 | 0.414 | 0.076 | 13 | 0.222 | 0.076 | 0.222 | 0.125 |
| | GATES | | 35.363 | 0.280 | 0.754 | 0.894 | 0.543 | 0.133 | 15 | 0.235 | 0.081 | 0.196 | 0.145 | 27.536 | 0.516 | 0.903 | 0.935 | 0.168 | 2.760 | 7 | 0.121 | 0.152 | 0.060 | 0.061 |
| | GREC | | 27.989 | 0.400 | 0.725 | 0.775 | 0.722 | 0.180 | 10 | 0.416 | 0.272 | 0.416 | 0.384 | 24.180 | 0.410 | 0.770 | 0.854 | 0.721 | 2.130 | 10 | 0.400 | 0.200 | 0.416 | 0.340 |
| | SEEC | | 16.004 | 0.542 | 0.885 | 0.913 | 0.855 | 0.325 | 16 | 0.421 | 0.152 | 0.210 | 0.355 | 10.041 | 0.633 | 0.936 | 0.953 | 0.802 | 2.066 | 11 | 0.422 | 0.286 | 0.351 | 0.338 |
| All | Llama-2 | 7B | 32.819 | 0.076 | 0.340 | 0.670 | 0.331 | 0.0045 | 107 | 0.069 | 0 | 0.069 | 0.139 | 26.950 | 0.102 | 0.357 | 0.659 | 0.331 | 0.0068 | 191 | 0.115 | 0.034 | 0.100 | 0.160 |
| | Llama-3 | 8B | 36.437 | 0.115 | 0.439 | 0.724 | 0.189 | 0.0046 | 95 | 0.015 | 0 | 0.015 | 0.069 | 24.242 | 0.084 | 0.281 | 0.575 | 0.414 | 0.0139 | 213 | 0.109 | 0.042 | 0.09 | 0.178 |
| | Claude | 4.5 | 35.769 | 0.052 | 0.329 | 0.677 | 0.474 | 0.0003 | 102 | 0.060 | 0 | 0.060 | 0.192 | 26.617 | 0.038 | 0.280 | 0.627 | 0.711 | 0.0032 | 142 | 0.086 | 0 | 0.086 | 0.151 |
| | Gemini | 2.5 | 39.962 | 0.035 | 0.227 | 0.542 | 0.268 | 0.0009 | 216 | 0.016 | 0 | 0.016 | 0.084 | 37.760 | 0.056 | 0.322 | 0.659 | 0.581 | 0.0007 | 218 | 0.033 | 0 | 0.026 | 0.065 |
| EMPATH (Relevant) | | medium | 14.567 | 0.796 | 0.933 | 0.937 | 0.802 | 0.338 | 10 | 0.390 | **0.291** | 0.390 | 0.358 | 11.762 | 0.784 | 0.942 | 0.942 | 0.790 | 2.125 | 8 | 0.400 | 0.333 | 0.400 | 0.325 |
| | | large | **12.231** | **0.800** | **0.948** | **0.942** | **0.964** | **0.371** | 7 | **0.500** | 0.253 | **0.416** | **0.454** | **10.008** | **0.788** | **0.958** | **0.961** | **0.826** | **3.343** | 7 | **0.500** | **0.374** | **0.500** | **0.345** |

Table 6: Qualitative analysis and Human A/B Testing against baselines. (*Note: IDK→ I Dont Know*)

| | Model | | Qualitative Analysis | | | | | | | | | | | | Human A/B Testing | | | | | | | | | | | |
|---|---|---|---|---|---|---|---|---|---|---|---|---|---|---|---|---|---|---|---|---|---|---|---|---|---|---|
| | | | Empathy(κ=0.88) Affective | Cognitive | Generic | IDK | Appropriateness(κ=0.82) Appropriate | Not A | IDK | Contextualization(κ=0.85) Contextual | Not C | IDK | Correctness(κ=0.89) Correct | Incorrect | IDK | E-T5 (M) W% | L% | T% | E-T5 (L) W% | L% | T% | EMPATH (M) W% | L% | T% | EMPATH (L) W% | L% | T% |
| No Emotion | GPT-2 small | 12 | 37 | 48 | 3 | 68 | 25 | 7 | 18 | 79 | 3 | 79 | 15 | 6 | 68.6 | 8.2 | 23.2 | 76.6 | 11.2 | 12.2 | 69.0 | 9.8 | 21.2 | 76.0 | 11.8 | 12.2 |
| | medium | 7 | 33 | 54 | 6 | 68 | 24 | 8 | 19 | 77 | 4 | 74 | 17 | 9 | 70.4 | 10.1 | 19.5 | 78.4 | 13.1 | 8.5 | 71.4 | 10.0 | 18.6 | 78.4 | 13.1 | 8.5 |
| | large | 5 | 41 | 47 | 7 | 64 | 29 | 7 | 18 | 78 | 4 | 75 | 20 | 5 | 68.1 | 17.9 | 14.0 | 76.1 | 20.9 | 3.0 | 69.7 | 18.4 | 11.9 | 76.7 | 20.3 | 3.0 |
| | DialoGPT small | 7 | 28 | 45 | 20 | 70 | 21 | 9 | 25 | 72 | 3 | 78 | 14 | 8 | 71.2 | 8.4 | 20.4 | 79.2 | 11.4 | 9.4 | 72.1 | 12.2 | 15.7 | 79.1 | 11.5 | 9.4 |
| | medium | 10 | 24 | 53 | 13 | 62 | 28 | 10 | 17 | 79 | 4 | 78 | 14 | 8 | 70.0 | 10.6 | 19.4 | 78.0 | 13.6 | 8.4 | 73.0 | 13.5 | 13.5 | 80.0 | 11.6 | 8.4 |
| | large | 3 | 31 | 50 | 16 | 60 | 25 | 15 | 16 | 80 | 4 | 77 | 18 | 5 | 60.9 | 25.8 | 13.3 | 68.9 | 28.8 | 2.3 | 71.6 | 13.9 | 14.5 | 78.6 | 19.1 | 2.3 |
| | GPT-Neo | 8 | 17 | 73 | 2 | 49 | 35 | 16 | 9 | 85 | 6 | 70 | 30 | 0 | 69.5 | 14.9 | 15.6 | 77.5 | 17.9 | 4.6 | 75.1 | 19.7 | 5.2 | 82.1 | 13.3 | 4.6 |
| | T5-base | 20 | 29 | 26 | 25 | 76 | 19 | 5 | 28 | 57 | 15 | 70 | 30 | 0 | 55.4 | 18.7 | 25.9 | 63.4 | 21.7 | 14.9 | 60.2 | 20.0 | 19.8 | 67.2 | 17.9 | 14.9 |
| | E-T5 medium | 62 | 18 | 15 | 5 | 57 | 23 | 20 | 20 | 50 | 30 | 90 | 10 | 0 | - | - | - | - | - | - | - | - | - | - | - | - |
| | large | 74 | 8 | 15 | 3 | 64 | 11 | 25 | 23 | 60 | 17 | 92 | 8 | 0 | - | - | - | - | - | - | - | - | - | - | - | - |
| 1 Emotion | SOTA EMMA | 12 | 25 | 53 | 10 | 77 | 20 | 3 | 22 | 73 | 5 | 91 | 7 | 2 | 67.4 | 23.1 | 9.5 | 75.4 | 18.8 | 5.8 | 69.2 | 21.6 | 9.2 | 76.2 | 20.0 | 3.8 |
| | ECTG | 14 | 34 | 35 | 17 | 76 | 10 | 14 | 19 | 70 | 11 | 91 | 6 | 3 | 59.8 | 10.6 | 29.6 | 67.8 | 13.6 | 18.6 | 61.1 | 10.3 | 28.6 | 68.1 | 13.3 | 18.6 |
| | GATES | 9 | 38 | 48 | 5 | 55 | 28 | 17 | 19 | 78 | 3 | 89 | 8 | 3 | 73.5 | 7.7 | 18.8 | 81.5 | 10.7 | 7.8 | 73.5 | 8.1 | 18.4 | 80.5 | 11.7 | 7.8 |
| | GREC | 17 | 27 | 45 | 11 | 64 | 16 | 20 | 28 | 71 | 1 | 87 | 9 | 4 | 61.3 | 4.0 | 34.7 | 69.3 | 14.0 | 16.7 | 65.1 | 7.1 | 27.8 | 72.1 | 11.2 | 16.7 |
| | SEEC | 29 | 48 | 5 | 18 | 80 | 11 | 9 | 58 | 40 | 2 | 90 | 8 | 2 | 43.0 | 10.0 | 47.0 | 51.0 | 13.0 | 36.0 | 49.8 | 11.5 | 38.7 | 56.8 | 7.2 | 36.0 |
| All | Llama-2 7B | 20 | 52 | 26 | 2 | 89 | 11 | 0 | 100 | 0 | 0 | 100 | 0 | 0 | 53.8 | 10.0 | 36.2 | 68.7 | 9.0 | 22.3 | 40.0 | 9.8 | 50.2 | 48.3 | 0 | 52.7 |
| | Llama-3 8B | 23 | 58 | 17 | 2 | 89 | 10 | 1 | 100 | 0 | 0 | 100 | 0 | 0 | 40.1 | 5.3 | 54.6 | 30.5 | 5.3 | 64.2 | 48.1 | 1.9 | 50.0 | 42.8 | 0 | 57.2 |
| | Claude 4.5 | 14 | 60 | 21 | 5 | 87 | 13 | 0 | 100 | 0 | 0 | 100 | 0 | 0 | 31.6 | 2.2 | 66.2 | 31.3 | 10.1 | 58.6 | 41.8 | 0 | 50.2 | 38.3 | 0 | 52.7 |
| | Gemini 2.5 | 27 | 63 | 9 | 1 | 86 | 14 | 0 | 100 | 0 | 0 | 100 | 0 | 0 | 51.1 | 3.6 | 45.3 | 45.8 | 6.8 | 47.4 | 45.5 | 5.0 | 49.5 | 48.8 | 0 | 51.2 |
| EMPATH (Relevant) | medium | 63 | 18 | 15 | 4 | 85 | 10 | 5 | 90 | 10 | 0 | 100 | 0 | 0 | - | - | - | - | - | - | - | - | - | - | - | - |
| | large | 75 | 11 | 11 | 3 | 87 | 6 | 7 | 95 | 5 | 0 | 100 | 0 | 0 | - | - | - | - | - | - | - | - | - | - | - | - |

## 5.5 RQ4: Importance of Multiple Emotions

To depict the effectiveness of our model with multiple relevant emotions, we compare it against all the cause-related ERG models and various generative models. Table 5 unfolds the improvements in various metrics for both empathetic dialogue datasets. Both E-T5 and EMPATH, fine-tuned on these datasets, surpass the current SOTA models incorporating emotional causes. Furthermore, by systematically removing or modifying specific elements, we aim to assess their impact on the overall performance of EMPATH. Results in Table 3 depict that the perplexity of EMPATH is sub-par with the removal of any component, but largely depends on the E-T5 model that incorporates affective empathy. Moreover, considering all emotions without their relevance also affects the performance and makes it similar to other LLMs.

## 6 CONCLUSION

In this work, we introduce a novel mechanism to jointly incorporate affective and cognitive empathy in dialogue agent responses by additional pre-training on a large-scale synthetic dataset. Incorporating this in EMPATH, we enhance ERG by addressing causes of multiple emotions of a conversation, overlooked by existing works. Extensive quantitative and qualitative experiments prove the efficiency of our dataset and framework in improving empathetic response generation. By introducing the dataset and EMPATH, we aim to significantly improve user experience (Appendix F) in Gen AI applications, making them more responsive and emotionally attuned to user needs. In the future, we wish to test the effectiveness of our dataset by pre-training other existing LLMs for enhanced ERG.

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

# A  APPENDIX

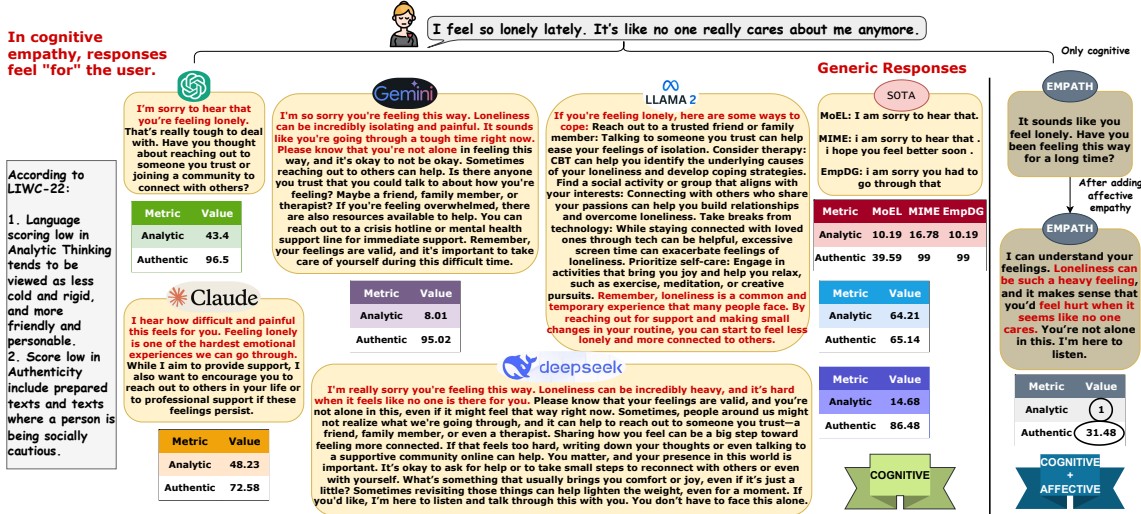

Figure 4: Analysis of responses generated by existing ERG works focused on cognitive empathy, using the LIWC-22 (Boyd et al., 2022) tool, reveals that EMPATH scores lowest in the considered metrics, depicting its friendly, personable, and socially cautious nature.

## LIMITATIONS

1. The dataset might still contain bias even after multiple checks. This is also a future direction where we try to mitigate this.

2. The golden response for all datasets had to be appropriately modified to evaluate the effectiveness of affective empathy.

3. The usage of causes is not position-dependent. For instance, if a cause of the sadness emotion is in the initial part of the conversation and one cause is more recent, then it's important to give higher importance to the recent one. This also enhances the context-relevance of the response.

4. The low-parameter models generate empathetic responses across most situations, i.e., whether empathy is appropriate or not (Appendix G). In contrast, the higher-parameter models appear largely unaffected by the 8.5GB dataset, suggesting that a substantially larger or more diverse corpus is required to cultivate affective empathy in them.

## ETHICAL CONSIDERATIONS

Our pre-trained generative model, EMPATH, aims to consistently respond empathetically, but the usage of this raw model might inadvertently validate behaviors, such as self-pity, without encouraging seeking professional help. Hence, it is advised to use it carefully, as increased reliance on the model for emotional support might lead to decreased real-world human interaction, which is crucial for mental and emotional well-being. All 6 human annotators employed in this work are research scholars who have received no payment for their contribution. We do not collect any personal information about them except their qualification and use the screening criteria mentioned in Appendix B. They were made well aware of the task at hand in advance.

## B    HUMAN EVALUATION DETAILS

**Selection of annotators:** We begin by choosing 10 research scholars for this task, well-versed in the English language. In order to help them understand empathy, we provide them with the works of renowned psychologists Dr. Brene Brown (Brown, 2018) and Dr. Marshal Rosenberg (Rosenberg & Chopra, 2015). In her book *Dare To Lead*, Section 4 titled Shame and Empathy, Dr. Brown mentions how empathy can be shown in different scenarios. She also details the various dos and don'ts of Empathy and how it can be differentiated from sympathy. The other work we refer to is Dr. Marshal Rosenberg's book, *Nonviolent Communication: A Language of Life*, Chapter 8: The Power of Empathy. Both these works quote extensive examples of how empathy can be shown to different people in various scenarios. Following this, we use the inventory defined by (McLaren, 2013) to test their empathetic knowledge. The inventory contains 44 statements and the individuals are supposed to answer with a 'yes' or 'no'. Based on the number of 'yes', scores are assigned and we select the top six scorers with over 75% in the assessment (33-44 yes as responses). To further enhance their knowledge of empathy, these six individuals are then provided with author and researcher, Karla McLaren's book, *The Art of Empathy: A Complete Guide to Life's Most Essential Skill*. The top three scorers are assigned as verifiers, and the next three scorers are assigned as annotators in our work.

Table 7: Evaluation criteria given to the human annotators in Section 5.4.

| Dimension | Question | Annotators' Decision | | Option Definition |
|---|---|---|---|---|
| | | *Value* | *Explanation Options* | |
| Empathy | *In the proposed response, how empathetic is the agent?* | Affective | "The proposed response is affectively empathetic and not generic." | Dialogue agent does not show sympathy to the speaker and does not address the emotions with generic replies. |
| | | | Add free form text explanation | |
| | | Cognitive | "The proposed response is cognitively empathetic and not generic." | Dialogue agent shows sympathy to the speaker and does not address the emotions with generic replies. |
| | | | Add free form text explanation | |
| | | Generic | "The proposed response is not completely empathetic but only shows generic emotional intent." | The response does not respond to the user with empathy. |
| | | | Add free form text explanation | |
| | | I don't know | Please Add free form text explanation (required) | The candidate contains some empathy but not enough. |
| Appropriateness | *Is the proposed response candidate appropriate?* | Appropriate | "The proposed response is coherent with the Dialogue context." | The response makes sense, and it can be the natural continuation of the shown dialogue context. |
| | | | Add free form text explanation | |
| | | Not Appropriate | "The proposed response is not coherent with the dialogue context." | The response does not make sense in the current dialogue context. |
| | | | Add free form text explanation | |
| | | I don't know | Please Add free form text explanation (required) | The candidate contains some elements that make sense to the dialogue context, but some that do not. |
| Contextualization | *Does the proposed response contain references to the context of the dialogue?* | Contextualized | "The response is referring to the dialogue context." | The candidate contains implicit or explicit references to the dialogue context. |
| | | | Add free form text explanation | |
| | | Not Contextualized | "The response is generic or does not contain any explicit or implicit reference to what it has been said in the dialogue context." | The candidate doesn't contain any reference to the dialogue context, or contains references that are incoherent with the dialogue context. |
| | | | "The response is not consistent with the information contained in the dialogue context." | |
| | | | Add free form text explanation | |
| | | I don't know | Please Add free form text explanation (required) | The response contains some references to the dialogue context, but contains other references that are not clear or relevant. |
| Correctness | *Is the proposed response grammatically correct?* | Correct | "The response is acceptable." | The response does not contain any type of grammatical or structural error, any repetitions, misspellings or any other types of error. |
| | | | Add free form text explanation | |
| | | Incorrect | "The response contains grammatical errors." "The response contains one or more parts that are repetitive." | The response contains some grammatical or structural errors such as, repetitions, misspelling, any other types of error. |
| | | | Add free form text explanation | |
| | | I don't know | Please Add free form text explanation (required) | It is hard to identify if the response contains errors or not. |

**Section 5.4 Qualitative Analysis**:- In our work, we compare our model on four different factors: Empathy, Appropriateness, Contextualization, and Correctness. To provide our annotators with the evaluation criteria, we follow the work of (Mousavi et al., 2022) and use their definition for three of these dimensions, i.e., Appropriateness, Contextualization, and Correctness. Similarly, define the Empathy factor, as shown in Table 7. In simple words, 1) Empathy helps determine the empathetic quotient of the predicted response, 2) Appropriateness helps determine the relevance of the response with the conversation history, 3) Contextualization helps figure out to what extent the response includes the history and is not generic and 4) Correctness

analyzes the fluency of the response. The annotators must answer the dimension question for each conversation using the assigned three option values and choose/provide an explanation for their option. For instance, when evaluating the Empathy factor for a conversation, the annotators must answer the question '*In the proposed response, how empathetic is the agent?*' with one of the three options, i.e., 'Empathetic', 'Generic' or 'I dont́ know' by referring to their definitions and provide an explanation for the chosen value. We decide the final verdict by performing a majority vote among the three results obtained for each conversation. A similar process must be followed for all considered dimensions.

**Section 5.4 Human A/B testing**:- Given two models, $X$ and $Y$— in our case, Baselines, and our models, anonymously — we ask the chosen annotators to pick the model with the best response for each sampled test instance. The annotators can select a Tie if the responses from both models seem equal. The final verdict on each instance is determined by majority voting. In case no two annotators agree on a selection, all three annotators reached three distinct conclusions: A, B, and Tie, we bring in a $4^{th}$ annotator. From this, we calculate the percentage of samples where A or B generates the better response and where A and B are equal.

## C  QUANTITATIVE METRICS

We use 2 different types of metrics for automatic evaluation of our work. The following are the **reference-free metrics**:

1. Perplexity: evaluates the predictive power of a language model on a sample of text. It is computed as the test set's inverse probability normalized by word count. Lower perplexity indicates better performance.

2. Distint-n (n=1,2,3): computes the ratio of unique n-grams to total n-grams, which indicates the diversity of generated replies. Higher values indicate more diverse responses.

3. Average Length: provides useful insights into the adequacy and conciseness of generated empathetic responses. Responses that are too short may fail to provide sufficient context or emotional support.

The following are the **reference-based metrics**:

1. BERTScore/Sentence Embedding Similarity: measures the semantic similarity between generated responses and reference responses using sentence embeddings. Cosine similarity is commonly used to compute similarity scores.

2. METEOR: helps ensure that models produce responses that are not only linguistically correct but also emotionally appropriate.

3. BLEU: The BLEU score can be used to evaluate the lexical similarity of generated empathetic responses to reference responses, focusing on n-gram precision and length. It counts the number of 1-grams in the generated response that match the 1-grams in the reference response.

4. ROUGE-N F1-Score (n=1,2,L): evaluates both linguistic and emotional accuracy of the empathetic response by providing a detailed analysis of n-gram overlap, sequence alignment, and overall balance between precision and recall.

## D  DATASET DETAILS

In Section 3, our main task is to generate a dataset large enough to additionally pre-train various LLMs so they can generate affectively empathetic statements. To do so, we follow three simple steps, as mapped

out in Figure 5. Upon generating initial statements from ChatGPT, we wish to paraphrase them using the Pegasus paraphraser, chosen based on the results in Table 8. In order to verify the number of statements the Pegasus paraphraser must generate at once, we experiment with 500 empathetic statements and record the number of good:bad sentences generated by the paraphraser. For instance, a statement like "You can count on me for support" is given to the paraphraser, and we record the number of good and bad paraphrased sentences generated when the limit is set from 10 and varied at intervals of 10. We vary this value till we obtain a value that simultaneously gives the optimal number of good statements and a minimal number of bad statements. Based on our findings, as reported in Figure 6, we conclude that $limit = 100$ gives the most number of paraphrased sentences with the least number of bad sentences to discard. We stop at 120 as we observe that the paraphraser fails and generates more unintelligible sentences after 100.

| Model | Distinct-n Scores | | | Self-BLEU |
|---|---|---|---|---|
| | $n = 1$ | $n = 2$ | $n = 3$ | |
| Pegasus | 0.588 | 0.926 | 0.929 | 0.851 |
| FLAN-T5 | 0.504 | 0.888 | 0.910 | 0.839 |
| GPT-4 | 0.529 | 0.886 | 0.917 | 0.767 |
| Gemini | 0.496 | 0.904 | 0.922 | 0.423 |
| Claude | 0.492 | 0.840 | 0.868 | 0.716 |

Table 8: Comparison of models for paraphrasing.

Figure 6: Performance of various paraphrase limits from $10 - 120$. *Blue indicates the number of good statements and red indicates the number of bad ones.*

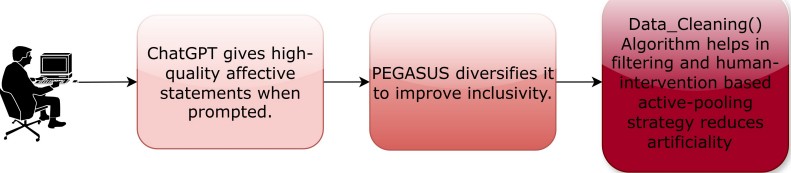

Figure 5: Caption

The bias check step in the process is carried out by Perspective, which recommends that comments with scores between $0.3$ and $0.7$ must be reviewed as the model is uncertain about them. Since our statements do not even cross $0.2$, we consider it as the threshold hereafter.

Table 9: Analysis results using the Perspective tool.

| Criteria | Bias Statistics | | | | | |
|---|---|---|---|---|---|---|
| | ChatGPT | | ChatGPT $+ level_1$ | | ChatGPT $+ level_1 + level_2$ | |
| | *Mean* | *Max* | *Mean* | *Max* | *Mean* | *Max* |
| Sexually Explicit | 0.01146 | 0.01346 | 0.00970 | 0.15888 | 0.01111 | 0.15968 |
| Toxicity | 0.02019 | 0.02584 | 0.01894 | 0.02584 | 0.02087 | 0.02584 |
| Identity Attack | 0.00384 | 0.00529 | 0.00357 | 0.00671 | 0.00374 | 0.00671 |
| Insult | 0.00902 | 0.01267 | 0.00899 | 0.01267 | 0.00955 | 0.01398 |
| Profanity | 0.01387 | 0.02019 | 0.01292 | 0.02019 | 0.01348 | 0.02188 |
| Threat | 0.00784 | 0.01512 | 0.00735 | 0.01512 | 0.00752 | 0.01512 |

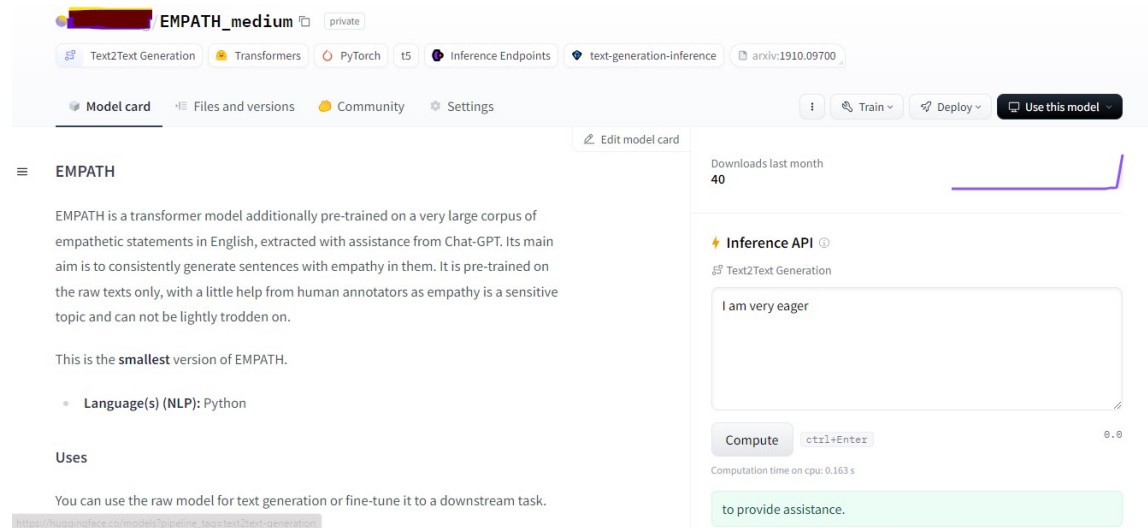

Figure 7: Deployment of EMPATH on HuggingFace.

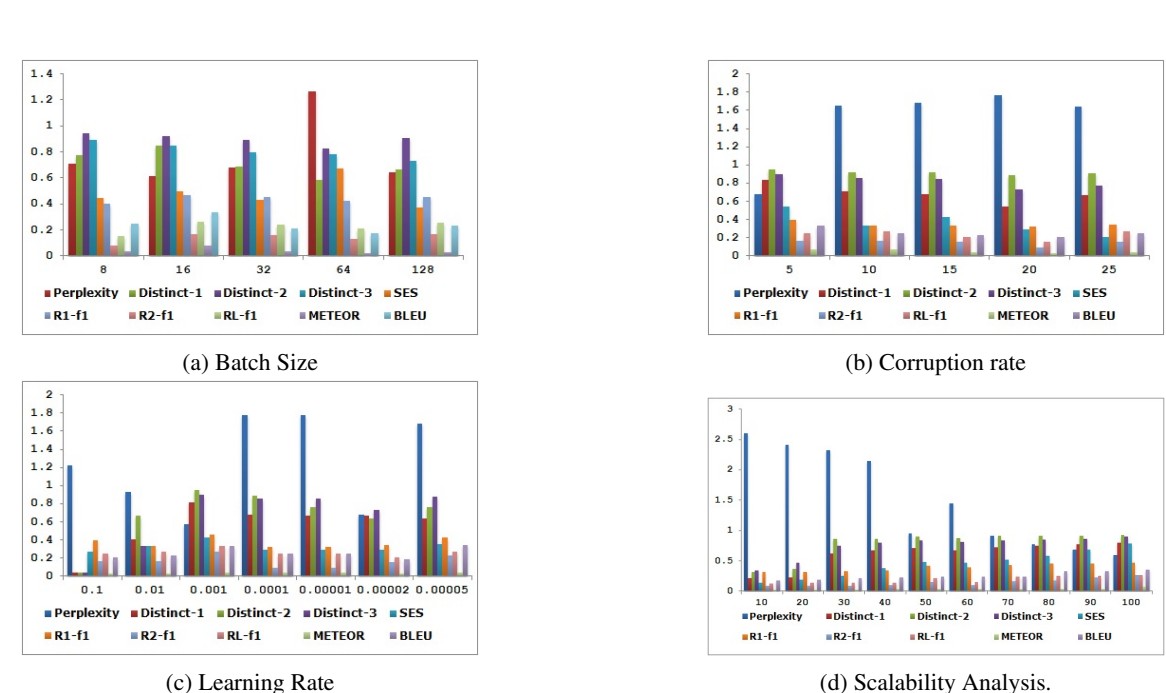

(a) Batch Size

(b) Corruption rate

(c) Learning Rate

(d) Scalability Analysis.

Figure 8: Hyper Parameter Tuning

# E  CASE STUDY

Our work focuses on presenting a generative LLM that helps generate affective empathetic responses based on the context. We show a case study where we consider a sample conversation and compare the replies generated by various models in Table 10.

| CONVERSATION EXAMPLE |
|---|
| ***Conversation History*** |
| Utterance 1: I am upset. |
| Utterance 2: Why? What happened? |
| Utterance 3: We broke up. |
| Utterance 4: On dear! it must be hurtful. Did you argue over something? |
| Utterance 5: No we did not and that idiot just went and told me its over. How dare he? |
| ***Generated Responses on fine-tuning using EmpatheticDialogue dataset*** |
| **EMMA**: I'm sorry for that. |
| **MEED2**: I am sorry you had to go through that. |
| **MoEL**: I am so sorry to hear that. Are you ok? |
| **MIME**: That is a bummer. What happened? |
| **SEEC**: I am sorry to have made you so angry. I was only trying to understand. |
| **EmpDG**: Oh no! I am so sorry. I am so sorry to hear that. I am so sorry to hear that. I am so sorry to hear that . |
| **GPT-2** *(small)*: You should be careful about that. |
| **GPT-2** *(medium)*: Oh yeah! I can understand. |
| **GPT-2** *(large)*: Yeah that is too bad. I don't really feel good about it. |
| **DialoGPT** *(small)*: I'm not sure. |
| **DialoGPT** *(medium)*: Do you have anything planned? |
| **DialoGPT** *(large)*: You should be careful with that when you get married. |
| **Llama-2** *(7B)*: Ugh, that sounds incredibly frustrating and hurtful. It's not fair when someone just ends things without any explanation or warning, especially when they didn't even give you a chance to defend yourself or work through any issues. It's totally understandable that you would be feeling upset and angry right now. It's like, you were just living your life and enjoying each other's company, and then BAM! Everything changes without any reason or warning. It's not fair to you, and it's not fair to the relationship. But remember, you don't have to go through this alone. There are people who care about you and want to support you, whether it's friends, family members, or even a therapist. Talking to someone about how you're feeling can really help you process everything and start to heal. And hey, maybe this is an opportunity for you to learn and grow as a person. Maybe you'll come out of this experience stronger and wiser, with a newfound appreciation for yourself and your worth. It won't be easy, but it could be worth it in the end. Just take care of yourself and be kind to yourself during this time. Remember to breathe, eat well, get enough sleep, and do things that make you happy. And don't forget to practice self-care – whether it's taking a relaxing bath, reading a book, or watching a funny movie – do whatever feels good for you and helps you unwind. You got this, girl! *hugs*. |
| **Llama-3** *(8B)*: It sounds like you're feeling really blindsided and betrayed by his sudden decision to break up with you without any warning or explanation. It's completely unacceptable for someone to treat you like that, especially if you were in a relationship with them. You deserved better than to be ghosted or dumped without any respect or compassion. Remember that his behavior has nothing to do with you and everything to do with his own insecurities and lack of emotional intelligence. You don't deserve to be treated like that, and you're better off without someone who can't communicate with you in a healthy way. Take some time to process your emotions and focus on yourself right now. You're strong, capable, and loved. Don't let someone else's poor behavior define your self-worth. You got this, and you'll come out of this even stronger on the other side. If you need someone to talk to or just need to vent, I'm here for you. Sending you lots of hugs and positive vibes! |
| **GPT-Neo**: That like what be wow for's when.,. you hang will the that your. |
| **T5-base**: I'm sorry for what happened. |
| **E-T5** *(medium)*: Your feelings matter about the break up. I am here to support you. |
| **E-T5** *(large)*: You can count on me for supporting you. |
| **EMPATH** *(medium)*: I am eager to know more about your feelings on this breakup. |
| **EMPATH** *(large)*: You can count on to be here. Please do tell me more about the break up. |

Table 10: A sample case study with responses generated by various models for the given conversation.

## F  STATISTICAL IMPORTANCE

In this Section, we assess whether the improvements of EMPATH over the baselines are statistically meaningful. To this end, we conduct a Wilcoxon signed-rank test McDonald & of Delaware (2009) across all evaluation metrics for the test set of conversations from respective datasets. Across both CHASE (500 conversations) and EmpatheticDialogue ($2.4k$ conversations), EMPATH demonstrates statistically significant improvements over all baseline models on nearly every metric, with Wilcoxon p-values frequently reaching extremely small magnitudes ($10^{-78}$–$10^{-165}$).

Table 11: Wilcoxon p-values of EMPATH against baselines across various metrics.

| Model | | PPL(↓) | Distinct(↑) | | | SES(↑) | BLEU(↑) | ROUGE F1-Score(↑) | | | METEOR(↑) |
|---|---|---|---|---|---|---|---|---|---|---|---|
| **CHASE** (test size = 500 conversations) | | | | | | | | | | | |
| | | | 1 | 2 | 3 | | | 1 | 2 | L | |
| GPT-2 | small | 2.54E-78 | 6.44E-68 | 6.05E-36 | 1.97E-64 | 6.32E-84 | 6.32E-84 | 6.32E-84 | 1.64E-78 | 6.32E-84 | 6.32E-84 |
| | medium | 8.28E-79 | 0.1466417 | 5.84E-36 | 3.82E-82 | 6.32E-84 | 2.19E-62 | 6.32E-84 | 6.32E-84 | 6.32E-84 | 6.32E-84 |
| | large | 1.24E-80 | 6.32E-84 | 1.26E-83 | 2.94E-17 | 6.32E-84 | 6.32E-84 | 6.32E-84 | 5.17E-82 | 6.32E-84 | 6.32E-84 |
| DialoGPT | small | 1.69E-80 | 6.32E-84 | 6.32E-84 | 2.43E-81 | 6.32E-84 | 6.32E-84 | 6.32E-84 | 4.32E-82 | 6.32E-84 | 6.32E-84 |
| | medium | 1.00E-47 | 6.32E-84 | 6.32E-84 | 2.94E-17 | 6.32E-84 | 6.32E-84 | 6.32E-84 | 9.59E-83 | 6.32E-84 | 6.32E-84 |
| | large | 4.32E-50 | 6.32E-84 | 6.32E-84 | 1.01E-45 | 6.32E-84 | 6.32E-84 | 6.32E-84 | 1.55E-71 | 6.32E-84 | 6.32E-84 |
| GPT-Neo | | 5.41E-79 | 6.32E-84 | 6.32E-84 | 2.49E-31 | 6.32E-84 | 6.32E-84 | 6.32E-84 | 6.32E-84 | 6.32E-84 | 6.32E-84 |
| T5-base | | 8.10E-42 | 6.32E-84 | 5.34E-76 | 7.85E-11 | 6.32E-84 | 6.32E-84 | 6.32E-84 | 6.32E-84 | 6.32E-84 | 6.32E-84 |
| SOTA | EMMA | 8.87E-83 | 6.32E-84 | 3.32E-64 | 0.0003342 | 6.32E-84 | 6.32E-84 | 6.32E-84 | 4.18E-83 | 6.32E-84 | 6.32E-84 |
| | ECTG | 6.32E-84 | 6.32E-84 | 0.0002231 | 1.57E-10 | 6.32E-84 | 6.32E-84 | 6.32E-84 | 1.02E-83 | 6.32E-84 | 6.32E-84 |
| | GATES | 6.32E-84 | 6.32E-84 | 6.32E-84 | 1.07E-65 | 6.32E-84 | 6.32E-84 | 6.32E-84 | 7.80E-84 | 6.32E-84 | 6.32E-84 |
| | GREC | 6.32E-84 | 6.32E-84 | 6.32E-84 | 6.32E-84 | 6.32E-84 | 6.32E-84 | 2.82E-79 | 2.66E-06 | 0.5379567 | 1.36E-70 |
| | SEEC | 7.04E-41 | 6.32E-84 | 4.37E-71 | 4.33E-43 | 6.36E-84 | 4.06E-78 | 7.46E-83 | 2.29E-77 | 6.32E-84 | 3.01E-82 |
| Llama-2 | 7B | 6.32E-84 | 6.32E-84 | 6.32E-84 | 6.32E-84 | 6.32E-84 | 6.32E-84 | 6.32E-84 | 6.32E-84 | 6.32E-84 | 6.32E-84 |
| Llama-3 | 8B | 6.32E-84 | 6.32E-84 | 6.32E-84 | 6.32E-84 | 6.32E-84 | 6.32E-84 | 6.32E-84 | 6.32E-84 | 6.32E-84 | 6.32E-84 |
| Claude | 4.5 | 6.32E-84 | 6.32E-84 | 6.32E-84 | 6.32E-84 | 6.32E-84 | 6.32E-84 | 6.32E-84 | 6.32E-84 | 6.32E-84 | 6.32E-84 |
| Gemini | 2.5 | 6.32E-84 | 6.32E-84 | 6.32E-84 | 6.32E-84 | 6.32E-84 | 6.32E-84 | 6.32E-84 | 6.32E-84 | 6.32E-84 | 6.32E-84 |
| **EmpatheticDialogue** (test size = 2.4k conversations) | | | | | | | | | | | |
| | | PPL(↓) | Distinct(↑) | | | SES(↑) | BLEU(↑) | ROUGE F1-Score(↑) | | | METEOR(↑) |
| | | | 1 | 2 | 3 | | | 1 | 2 | L | |
| GPT-2 | small | 2.55E-155 | 4.87E-159 | 5.17E-41 | 1.97E-163 | 1.66E-165 | 1.66E-165 | 1.66E-165 | 1.66E-165 | 1.66E-165 | 1.66E-165 |
| | medium | 1.66E-165 | 1.66E-165 | 1.06E-164 | 0.0020143 | 1.66E-165 | 1.66E-165 | 1.66E-165 | 1.66E-165 | 1.66E-165 | 1.66E-165 |
| | large | 1.66E-165 | 1.66E-165 | 1.66E-165 | 2.52E-165 | 1.66E-165 | 1.66E-165 | 1.66E-165 | 1.66E-165 | 1.66E-165 | 1.66E-165 |
| DialoGPT | small | 1.89E-125 | 1.66E-165 | 3.25E-102 | 0.1787781 | 1.66E-165 | 1.66E-165 | 1.66E-165 | 1.66E-165 | 1.66E-165 | 1.66E-165 |
| | medium | 4.80E-143 | 1.66E-165 | 2.80E-163 | 9.97E-69 | 1.66E-165 | 1.66E-165 | 1.66E-165 | 1.66E-165 | 1.66E-165 | 1.66E-165 |
| | large | 1.66E-165 | 2.69E-145 | 4.19E-27 | 5.54E-162 | 1.66E-165 | 1.66E-165 | 1.66E-165 | 1.66E-165 | 1.66E-165 | 1.66E-165 |
| GPT-Neo | | 1.66E-165 | 1.66E-165 | 1.66E-165 | 1.66E-165 | 1.66E-165 | 1.66E-165 | 1.66E-165 | 1.66E-165 | 1.66E-165 | 1.66E-165 |
| T5-base | | 1.48E-31 | 1.66E-165 | 1.66E-165 | 3.23E-164 | 1.66E-165 | 1.66E-165 | 1.66E-165 | 1.66E-165 | 1.66E-165 | 1.66E-165 |
| SOTA | EMMA | 1.66E-165 | 1.66E-165 | 1.66E-165 | 6.55E-165 | 1.66E-165 | 1.66E-165 | 1.66E-165 | 1.66E-165 | 1.66E-165 | 1.66E-165 |
| | ECTG | 1.66E-165 | 1.66E-165 | 1.66E-165 | 9.38E-134 | 1.66E-165 | 1.66E-165 | 1.66E-165 | 1.66E-165 | 1.66E-165 | 1.66E-165 |
| | GATES | 1.66E-165 | 1.66E-165 | 2.17E-144 | 4.76E-79 | 1.66E-165 | 3.33E-139 | 1.66E-165 | 1.66E-165 | 1.66E-165 | 1.66E-165 |
| | GREC | 1.66E-165 | 1.66E-165 | 1.66E-165 | 2.50E-165 | 4.99E-165 | 2.54E-165 | 9.59E-164 | 1.66E-165 | 3.09E-162 | 0.0014499 |
| | SEEC | 0.3810731 | 1.66E-165 | 6.37E-53 | 2.61E-14 | 7.63E-55 | 2.54E-165 | 3.21E-164 | 2.75E-157 | 1.66E-165 | 1.66E-165 |
| Llama-2 | 7B | 1.66E-165 | 1.66E-165 | 1.66E-165 | 1.66E-165 | 1.66E-165 | 1.66E-165 | 1.66E-165 | 1.66E-165 | 1.66E-165 | 1.66E-165 |
| Llama-3 | 8B | 1.66E-165 | 1.66E-165 | 1.66E-165 | 1.66E-165 | 1.66E-165 | 1.66E-165 | 1.66E-165 | 1.66E-165 | 1.66E-165 | 1.66E-165 |
| Claude | 4.5 | 1.66E-165 | 1.66E-165 | 1.66E-165 | 1.66E-165 | 1.71E-165 | 1.66E-165 | 1.66E-165 | 1.66E-165 | 1.66E-165 | 1.66E-165 |
| Gemini | 2.5 | 1.66E-165 | 1.66E-165 | 1.66E-165 | 1.66E-165 | 1.66E-165 | 1.66E-165 | 1.66E-165 | 1.66E-165 | 1.66E-165 | 1.66E-165 |

## G  EMPATHY BY LLMS UNDER VARIED CIRCUMSTANCES

In this section, we examine how various models behave under varied situations. Initially, we compare how various models perform when they are additionally pre-trained on our synthetic dataset. Due to the relatively small parameter sizes of many of these models, we observe that their behavior shifts dramatically: they begin responding over-empathetically in nearly all situations, failing to recognize when empathy is contextually appropriate. This aligns with the observations discussed in the Limitations section of this work. We also note that larger-sized LLMs like Llama-2, GPT-2 large, etc, do not show any behavior change, hence we do not report them in Table 1. A potential workaround is to carefully **control the scale of additional pre-training** or **to fine-tune the models on real conversational data** to help them learn when empathy is warranted and when it is not. Interestingly, we find that T5-base benefits from our synthetic dataset without exhibiting this over-generalization. The resulting model, E-T5, demonstrates an improved ability to regulate its empathetic responses, which is one of the key reasons we selected T5 as the foundation for our approach.

Table 12: Comparison of how models respond, when additionally pre-trained on affective empathy vs E-T5.

| | |
|---|---|
| User: Hey! you wont believe what happened today!
GPT-2 (small): I am here to support you emotionally through whatever you are going through.
User: I won the chess tournament!
GPT-2 (small): I understand what you are going through. I am here to support you in any way possible. | User: Hey! you wont believe what happened today!
E-T5: I am here to talk about what happened today.

User: I won the chess tournament!
E-T5: It's great that you won! I am here if you want to talk more about it. |
| User: I misplaced my house keys today.
GPT-2 (medium): I know losing something important can be so emotional.
User: I found them later in my jacket pocket.
GPT-2 (medium): I understand how hard that must have been for you. I'm here to support your feelings through this difficult moment. | User: I misplaced my house keys today.
E-T5: Did you find the house keys?

User: I found them later in my jacket pocket.
E-T5: Good to hear you found them. |
| User: Ugh, I accidentally burned my dinner tonight.
DialoGPT (small): I'm so sorry you had to experience that. I'm here to comfort you.
User: Just some pasta... it stuck to the bottom so I ate the unburnt part.
DialoGPT (small): I completely understand the frustration. I want you to know you're not alone in this. | User: Ugh, I accidentally burned my dinner tonight.
E-T5: I am sorry about the cooking.

User: Just some pasta... it stuck to the bottom so I ate the unburnt part.
E-T5: I am sorry you had to eat only the unburnt part. |
| User: I finally finished reading that novel I started.
DialoGPT (medium): I am here to discuss the novel with you.
User: It was a bit surprising but good overall.
DialoGPT (medium): I completely understand the emotional weight of surprising endings. You don't have to go through those feelings alone. | User: I finally finished reading that novel I started.
E-T5: Oh. How was the novel?
User: It was a bit surprising but good overall.
E-T5: Looks like you had a good read. |
| User: I misplaced my house keys today.
Llama-2 (w/o pre-training): Aww, sorry to hear you are going through that! Losing something as important as house keys can be really frustrating. Have you retraced your steps to see if you can find them? Sometimes, they might have fallen into a hidden crevice or underneath something where you didn't notice. If you're still unable to locate them, don't worry! Many locksmiths offer key duplication services, so you can get new ones made without having to break into your own home. Just give them a call and let them know what happened. They should be able to help you out. In the meantime, try to stay calm and think clearly. It's easy to panic when we lose something essential like this, but taking deep breaths and focusing on the solution can help us solve the problem faster. Good luck! | User: I misplaced my house keys today.
Llama-2 (pre-trained): Aww, sorry to hear you are going through that! Losing something as important as house keys can be really frustrating. Have you retraced your steps to see if you can find them? Sometimes, they might have fallen into a hidden crevice or underneath something where you didn't notice. If you're still unable to locate them, don't worry! Many locksmiths offer key duplication services, so you can get new ones made without having to break into your own home. Just give them a call and let them know what happened. They should be able to help you out. In the meantime, try to stay calm and think clearly. It's easy to panic when we lose something essential like this, but taking deep breaths and focusing on the solution can help us solve the problem faster. Good luck! |

Following this, we evaluate how frontier LLMs perform without any pre-training or fine-tuning, using only prompting. We designed a task-specific prompt (Table 13) that defines the task assigned to the LLM. Models were evaluated under 1) 0-shot: Only task instructions provided, with no in-context examples. 2) $1-$shot$/5-$shot: One or five in-context annotated examples were prepended before the test dialogue to guide generation.

Table 13: Prompt for $0-$shot inference in LLMs.

---

**Prompts for LLMs**

---

Generate an empathetic response to the following conversation with purely affective empathy and no cognitive empathy.
**Task**: Reply as $D$ for the conversation

- - - - - - - - - - - - - - - - - - - - - - - - - - - - - - - - - - - - - - -

**Conversation between Speaker $A$ and Dialogue Agent $D$:**
*A: I am upset.*
*D: Why? What happened?*
*A: We broke up.*
*D: Oh dear! It must be hurtful. Did you argue over something?*
*A: No we did not. That idiot just went and told me it's over. How dare he?*

- - - - - - - - - - - - - - - - - - - - - - - - - - - - - - - - - - - - - - -

**Output Requirements:**
1. Output the response.
2. Do not provide any explanation. Just the response.

---

As shown in Table 14, model behavior changes only minimally when prompted with the examples carefully selected by our annotators to demonstrate clear affective empathy. *Notably, we do not include ChatGPT in this comparison, as it serves both as the LLM-as-judge and as the model used to generate the dataset. Moreover, as DeepSeek, BART, and FLAN-T5 demonstrate patterns comparable to those of Gemini, they are excluded from our primary results.*

Table 14: Comparison of models by prompting using high temperatures.

**TEMPERATURE = 0.8**

| | | CHASE (test size = 500 conversations) | | | | | | | | | | | EmpatheticDialogue (test size = 2.4k conversations) | | | | | | | | | | |
|---|---|---|---|---|---|---|---|---|---|---|---|---|---|---|---|---|---|---|---|---|---|---|---|
| Model | | PPL(↓) | Distinct(↑) 1 | 2 | 3 | SES(↑) | BLEU(↑) | AL(↑) | ROUGE F1 1 | 2 | L | METEOR(↑) | PPL(↓) | Distinct(↑) 1 | 2 | 3 | SES(↑) | BLEU(↑) | AL(↑) | ROUGE F1 1 | 2 | L | METEOR(↑) |
| **Zero-shot** | | | | | | | | | | | | | | | | | | | | | | | |
| Llama-2 | 7B | 32.819 | 0.076 | 0.34 | 0.67 | 0.331 | 0.0045 | 107 | 0.069 | 0 | 0.069 | 0.139 | 26.95 | 0.102 | 0.357 | 0.659 | 0.331 | 0.0068 | 191 | 0.115 | 0.034 | 0.1 | 0.16 |
| Llama-3 | 8B | 36.437 | 0.115 | 0.439 | 0.724 | 0.189 | 0.0046 | 95 | 0.015 | 0 | 0.015 | 0.069 | 24.242 | 0.084 | 0.281 | 0.575 | 0.414 | 0.0139 | 213 | 0.109 | 0.042 | 0.09 | 0.178 |
| Claude | 4.5 | 35.769 | 0.052 | 0.329 | 0.677 | 0.474 | 0.0003 | 102 | 0.06 | 0 | 0.06 | 0.192 | 26.617 | 0.038 | 0.28 | 0.627 | 0.711 | 0.0032 | 142 | 0.086 | 0 | 0.086 | 0.151 |
| Gemini | 2.5 | 39.962 | 0.035 | 0.227 | 0.542 | 0.268 | 0.0009 | 216 | 0.016 | 0 | 0.016 | 0.084 | 37.76 | 0.056 | 0.322 | 0.659 | 0.581 | 0.0007 | 218 | 0.033 | 0 | 0.026 | 0.065 |
| FLAN-T5 | base | 39.681 | 0.035 | 0.227 | 0.542 | 0.268 | 0.0009 | 216 | 0.016 | 0 | 0.016 | 0.078 | 37.68 | 0.061 | 0.325 | 0.691 | 0.576 | 0.0006 | 220 | 0.033 | 0 | 0.026 | 0.058 |
| BART | large | 39.765 | 0.047 | 0.269 | 0.543 | 0.258 | 0.0009 | 235 | 0.016 | 0 | 0.016 | 0.081 | 37.72 | 0.06 | 0.32 | 0.666 | 0.59 | 0.0011 | 216 | 0.033 | 0 | 0.026 | 0.061 |
| Deepseek | 3.2 | 39.801 | 0.039 | 0.217 | 0.539 | 0.262 | 0.0009 | 222 | 0.016 | 0 | 0.016 | 0.082 | 37.74 | 0.058 | 0.323 | 0.684 | 0.581 | 0.0008 | 201 | 0.086 | 0 | 0.086 | 0.062 |
| **One-shot** | | | | | | | | | | | | | | | | | | | | | | | |
| Llama-2 | 7B | 29.709 | 0.098 | 0.368 | 0.681 | 0.35 | 0.0045 | 107 | 0.069 | 0 | 0.069 | 0.156 | 25.723 | 0.119 | 0.363 | 0.683 | 0.357 | 0.0068 | 191 | 0.115 | 0.034 | 0.1 | 0.184 |
| Llama-3 | 8B | 33.327 | 0.137 | 0.467 | 0.735 | 0.208 | 0.0046 | 95 | 0.015 | 0 | 0.015 | 0.086 | 23.015 | 0.101 | 0.287 | 0.599 | 0.44 | 0.0139 | 213 | 0.109 | 0.042 | 0.09 | 0.202 |
| Claude | 4.5 | 32.659 | 0.074 | 0.357 | 0.688 | 0.493 | 0.0003 | 102 | 0.06 | 0 | 0.06 | 0.209 | 25.39 | 0.055 | 0.286 | 0.651 | 0.737 | 0.0032 | 142 | 0.086 | 0 | 0.086 | 0.175 |
| Gemini | 2.5 | 37.42 | 0.052 | 0.238 | 0.57 | 0.302 | 0.0009 | 216 | 0.016 | 0 | 0.016 | 0.11 | 35.433 | 0.08 | 0.328 | 0.676 | 0.607 | 0.0007 | 218 | 0.033 | 0 | 0.026 | 0.089 |
| FLAN-T5 | base | 37.139 | 0.052 | 0.238 | 0.57 | 0.302 | 0.0009 | 216 | 0.016 | 0 | 0.016 | 0.104 | 35.353 | 0.085 | 0.331 | 0.708 | 0.602 | 0.0006 | 220 | 0.033 | 0 | 0.026 | 0.082 |
| BART | large | 37.223 | 0.064 | 0.28 | 0.571 | 0.292 | 0.0009 | 235 | 0.016 | 0 | 0.016 | 0.107 | 35.393 | 0.084 | 0.326 | 0.683 | 0.616 | 0.0011 | 216 | 0.033 | 0 | 0.026 | 0.085 |
| Deepseek | 3.2 | 37.259 | 0.056 | 0.228 | 0.567 | 0.296 | 0.0009 | 222 | 0.016 | 0 | 0.016 | 0.108 | 35.413 | 0.082 | 0.329 | 0.701 | 0.607 | 0.0008 | 201 | 0.086 | 0 | 0.086 | 0.086 |
| **Few(5)-shot** | | | | | | | | | | | | | | | | | | | | | | | |
| Llama-2 | 7B | 27.618 | 0.093 | 0.351 | 0.698 | 0.365 | 0.0045 | 107 | 0.069 | 0 | 0.069 | 0.165 | 26.623 | 0.126 | 0.383 | 0.676 | 0.337 | 0.0068 | 191 | 0.115 | 0.034 | 0.1 | 0.186 |
| Llama-3 | 8B | 31.236 | 0.132 | 0.45 | 0.752 | 0.223 | 0.0046 | 95 | 0.015 | 0 | 0.015 | 0.095 | 24.915 | 0.108 | 0.307 | 0.592 | 0.42 | 0.0139 | 213 | 0.109 | 0.042 | 0.09 | 0.204 |
| Claude | 4.5 | 30.568 | 0.069 | 0.34 | 0.705 | 0.508 | 0.0003 | 102 | 0.06 | 0 | 0.06 | 0.218 | 26.29 | 0.062 | 0.306 | 0.644 | 0.717 | 0.0032 | 142 | 0.086 | 0 | 0.086 | 0.177 |
| Gemini | 2.5 | 34.31 | 0.057 | 0.255 | 0.553 | 0.287 | 0.0009 | 216 | 0.016 | 0 | 0.016 | 0.101 | 36.533 | 0.073 | 0.348 | 0.683 | 0.587 | 0.0007 | 218 | 0.033 | 0 | 0.026 | 0.091 |
| FLAN-T5 | base | 34.029 | 0.057 | 0.255 | 0.553 | 0.287 | 0.0009 | 216 | 0.016 | 0 | 0.016 | 0.101 | 36.453 | 0.078 | 0.351 | 0.715 | 0.59 | 0.0006 | 220 | 0.033 | 0 | 0.026 | 0.084 |
| BART | large | 34.113 | 0.057 | 0.255 | 0.553 | 0.287 | 0.0009 | 216 | 0.016 | 0 | 0.016 | 0.101 | 36.493 | 0.077 | 0.346 | 0.69 | 0.596 | 0.0011 | 216 | 0.033 | 0 | 0.026 | 0.087 |
| Deepseek | 3.2 | 34.149 | 0.057 | 0.255 | 0.553 | 0.287 | 0.0009 | 216 | 0.016 | 0 | 0.016 | 0.101 | 36.513 | 0.075 | 0.349 | 0.708 | 0.587 | 0.0008 | 201 | 0.086 | 0 | 0.086 | 0.088 |

**TEMPERATURE = 0.9 (high)**

| | | CHASE | | | | | | | | | | | EmpatheticDialogue | | | | | | | | | | |
|---|---|---|---|---|---|---|---|---|---|---|---|---|---|---|---|---|---|---|---|---|---|---|---|
| Model | | PPL | D1 | D2 | D3 | SES | BLEU | AL | R1 | R2 | RL | METEOR | PPL | D1 | D2 | D3 | SES | BLEU | AL | R1 | R2 | RL | METEOR |
| **Zero-shot** | | | | | | | | | | | | | | | | | | | | | | | |
| Llama-2 | 7B | 32.819 | 0.089 | 0.348 | 0.683 | 0.331 | 0.0045 | 107 | 0.069 | 0 | 0.069 | 0.147 | 26.958 | 0.125 | 0.371 | 0.682 | 0.345 | 0.0068 | 191 | 0.115 | 0.034 | 0.1 | 0.168 |
| Llama-3 | 8B | 36.437 | 0.128 | 0.447 | 0.737 | 0.189 | 0.0046 | 95 | 0.015 | 0 | 0.015 | 0.077 | 24.25 | 0.107 | 0.295 | 0.598 | 0.428 | 0.0139 | 213 | 0.109 | 0.042 | 0.09 | 0.186 |
| Claude | 4.5 | 35.769 | 0.065 | 0.337 | 0.69 | 0.474 | 0.0003 | 102 | 0.06 | 0 | 0.06 | 0.2 | 26.625 | 0.061 | 0.294 | 0.65 | 0.725 | 0.0032 | 142 | 0.086 | 0 | 0.086 | 0.159 |
| Gemini | 2.5 | 39.962 | 0.048 | 0.235 | 0.555 | 0.268 | 0.0009 | 216 | 0.016 | 0 | 0.016 | 0.092 | 37.768 | 0.079 | 0.336 | 0.682 | 0.595 | 0.0007 | 218 | 0.033 | 0 | 0.026 | 0.073 |
| FLAN-T5 | base | 39.681 | 0.035 | 0.235 | 0.555 | 0.268 | 0.0009 | 216 | 0.016 | 0 | 0.016 | 0.086 | 37.688 | 0.084 | 0.339 | 0.714 | 0.59 | 0.0006 | 220 | 0.033 | 0 | 0.026 | 0.066 |
| BART | large | 39.765 | 0.06 | 0.277 | 0.556 | 0.258 | 0.0009 | 235 | 0.016 | 0 | 0.016 | 0.089 | 37.728 | 0.083 | 0.334 | 0.689 | 0.604 | 0.0011 | 216 | 0.033 | 0 | 0.026 | 0.069 |
| Deepseek | 3.2 | 39.801 | 0.052 | 0.225 | 0.552 | 0.262 | 0.0009 | 222 | 0.016 | 0 | 0.016 | 0.09 | 37.748 | 0.081 | 0.337 | 0.707 | 0.595 | 0.0008 | 201 | 0.086 | 0 | 0.086 | 0.07 |
| **One-shot** | | | | | | | | | | | | | | | | | | | | | | | |
| Llama-2 | 7B | 29.709 | 0.111 | 0.376 | 0.694 | 0.35 | 0.0045 | 107 | 0.069 | 0 | 0.069 | 0.164 | 25.731 | 0.142 | 0.377 | 0.706 | 0.371 | 0.0068 | 191 | 0.115 | 0.034 | 0.1 | 0.192 |
| Llama-3 | 8B | 33.327 | 0.15 | 0.475 | 0.748 | 0.208 | 0.0046 | 95 | 0.015 | 0 | 0.015 | 0.094 | 23.023 | 0.124 | 0.301 | 0.622 | 0.454 | 0.0139 | 213 | 0.109 | 0.042 | 0.09 | 0.21 |
| Claude | 4.5 | 32.659 | 0.087 | 0.365 | 0.701 | 0.493 | 0.0003 | 102 | 0.06 | 0 | 0.06 | 0.217 | 25.398 | 0.078 | 0.3 | 0.674 | 0.751 | 0.0032 | 142 | 0.086 | 0 | 0.086 | 0.183 |
| Gemini | 2.5 | 37.42 | 0.065 | 0.246 | 0.583 | 0.302 | 0.0009 | 216 | 0.016 | 0 | 0.016 | 0.118 | 35.441 | 0.103 | 0.342 | 0.699 | 0.621 | 0.0007 | 218 | 0.033 | 0 | 0.026 | 0.097 |
| FLAN-T5 | base | 37.139 | 0.065 | 0.246 | 0.583 | 0.302 | 0.0009 | 216 | 0.016 | 0 | 0.016 | 0.112 | 35.361 | 0.108 | 0.345 | 0.731 | 0.616 | 0.0006 | 220 | 0.033 | 0 | 0.026 | 0.09 |
| BART | large | 37.223 | 0.077 | 0.288 | 0.584 | 0.292 | 0.0009 | 235 | 0.016 | 0 | 0.016 | 0.115 | 35.401 | 0.107 | 0.34 | 0.706 | 0.63 | 0.0011 | 216 | 0.033 | 0 | 0.026 | 0.093 |
| Deepseek | 3.2 | 37.259 | 0.069 | 0.236 | 0.58 | 0.296 | 0.0009 | 222 | 0.016 | 0 | 0.016 | 0.116 | 35.421 | 0.105 | 0.343 | 0.724 | 0.621 | 0.0008 | 201 | 0.086 | 0 | 0.086 | 0.094 |
| **Few(5)-shot** | | | | | | | | | | | | | | | | | | | | | | | |
| Llama-2 | 7B | 27.618 | 0.106 | 0.359 | 0.711 | 0.365 | 0.0045 | 107 | 0.069 | 0 | 0.069 | 0.173 | 26.631 | 0.149 | 0.397 | 0.699 | 0.351 | 0.0068 | 191 | 0.115 | 0.034 | 0.1 | 0.194 |
| Llama-3 | 8B | 31.236 | 0.145 | 0.458 | 0.765 | 0.223 | 0.0046 | 95 | 0.015 | 0 | 0.015 | 0.103 | 24.923 | 0.131 | 0.321 | 0.615 | 0.434 | 0.0139 | 213 | 0.109 | 0.042 | 0.09 | 0.212 |
| Claude | 4.5 | 30.568 | 0.082 | 0.348 | 0.718 | 0.508 | 0.0003 | 102 | 0.06 | 0 | 0.06 | 0.226 | 26.298 | 0.085 | 0.32 | 0.667 | 0.731 | 0.0032 | 142 | 0.086 | 0 | 0.086 | 0.185 |
| Gemini | 2.5 | 34.31 | 0.07 | 0.263 | 0.566 | 0.287 | 0.0009 | 216 | 0.016 | 0 | 0.016 | 0.109 | 36.541 | 0.096 | 0.362 | 0.706 | 0.601 | 0.0007 | 218 | 0.033 | 0 | 0.026 | 0.099 |
| FLAN-T5 | base | 34.029 | 0.07 | 0.263 | 0.566 | 0.287 | 0.0009 | 216 | 0.016 | 0 | 0.016 | 0.109 | 36.461 | 0.101 | 0.365 | 0.738 | 0.596 | 0.0006 | 220 | 0.033 | 0 | 0.026 | 0.092 |
| BART | large | 34.113 | 0.07 | 0.263 | 0.566 | 0.287 | 0.0009 | 216 | 0.016 | 0 | 0.016 | 0.109 | 36.501 | 0.1 | 0.36 | 0.713 | 0.61 | 0.0011 | 216 | 0.033 | 0 | 0.026 | 0.095 |
| Deepseek | 3.2 | 34.149 | 0.07 | 0.263 | 0.566 | 0.287 | 0.0009 | 216 | 0.016 | 0 | 0.016 | 0.109 | 36.521 | 0.098 | 0.363 | 0.731 | 0.601 | 0.0008 | 201 | 0.086 | 0 | 0.086 | 0.096 |

