# OpenReview forum: "Be Affective, Not Just Cognitive - Towards Imparting Pertinent Empathy in Dialogue Agents"
_ICLR.cc/2026/Conference — Submitted to ICLR 2026_

### Official Review · Reviewer_NJTA · 2025-10-26

**Soundness:** 2
**Presentation:** 3
**Contribution:** 3
**Rating:** 6
**Confidence:** 4

**Summary:**

EMPATH effectively enhances affective empathy and response specificity through its synthetic dataset and emotion-aware loss design, achieving significant performance gains across several benchmarks. The explainability-based emotion cause extraction improves interpretability and conciseness compared to traditional clause-based methods. However, the paper lacks verification of the synthetic dataset’s generalization to real-world human dialogues, and the hyperparameter optimization process for the proposed loss remains insufficiently detailed. Furthermore, the statistical significance of reported improvements is not analyzed, and reproducibility information is incomplete.

**Strengths:**

1.The study introduces a large, well-curated synthetic dataset that substantially expands resources for empathetic dialogue modeling and effectively supports the learning of affective empathy.

2.The explainability-based emotion cause extraction improves interpretability and user relatability by achieving higher conciseness and accuracy than existing approaches.

3.The proposed emotion-aware contextual loss successfully reduces generic responses by balancing emotional alignment, contextual relevance, and empathy validation.

**Weaknesses:**

1.The real-world validity of the synthetic dataset remains untested. No experiments compare the model’s performance on synthetic versus human-written empathetic data, which raises concerns about its generalization to real user conversations.

2.The paper does not include statistical significance analysis of the results. All metrics are presented as single values without variance or confidence intervals, leaving uncertainty about whether the reported improvements over baselines are meaningful.

3.The process for tuning hyperparameters (λ₁/λ₂/λ₃) in the emotion-aware contextual loss is insufficiently described. The paper does not explain how the loss weights are optimized and lacks ablation studies to assess the contribution of each component.

4.The framework’s behavior in non-empathetic contexts is not examined. There is no evaluation of whether EMPATH can avoid expressing empathy when it is inappropriate, which may lead to misalignment with user intent.

5.The study does not consider multilingual or cross-cultural dimensions of empathy. All experiments are conducted in English, with no analysis of how cultural differences in emotional expression may influence model performance.

**Questions:**

See Weaknesses.

---

> ### Author Response · Authors · 2025-11-18
> **Response to Reviewer**
>
> Thank you so much for your positive response to our work. Based on your pointed out weaknesses, we have the following justification and amendments. We apologize for the delay caused due to conducting a few experiments.
> 1. $\textbf{Multi-lingual aspect}$: Currently, our work focuses exclusively on English. We agree that exploring multilingual and cross-cultural dimensions of empathy is an important future direction. In principle, large-scale empathetic data in other languages could be generated using LLMs like ChatGPT, as we do for English. However, current paraphrasing tools for other languages are limited, making large-scale dataset creation challenging at this stage.
> 2. $\textbf{Loss function}$: We have clarified this in the manuscript: "These hyperparameters ($\lambda_1, \lambda_2, \lambda_3$) control the relative importance of each component in the total loss. During optimization, they are updated jointly with the model parameters using the Adam optimizer, based on the gradients of the total loss with respect to each $\lambda$." Additionally, the ablation study is reported in Table 3 of the manuscript, showing the model’s performance when each loss component is removed, demonstrating the contribution of each term.
> 3. $\textbf{Behavior in non-empathetic context}$: We chose T5 for EMPATH because its parameter size helps retain the model’s basic training, including the ability to avoid empathy when inappropriate. In contrast, low-parameter models fine-tuned on the same dataset tend to lose this understanding and respond empathetically regardless of context. To illustrate this behavior, several sample responses have been added to the Appendix.
> 4. $\textbf{Real world validity}$: We appreciate this important point. We have addressed it in Tables 4 and 5, where, after pre-training on affective empathy, the model is fine-tuned on datasets containing real user conversations. Across all such datasets, our model consistently outperforms the baselines. Human evaluation also supports these findings, indicating that the synthetic pre-training transfers effectively to real-world dialogues. We are happy to consider additional experiments if there are specific tests the reviewers would recommend to further validate real-world generalization.
> 5. $\textbf{Statistical significance}$: For the human evaluation, we report inter-rater reliability using Fleiss’ Kappa, obtaining values of 0.88 for empathy, 0.82 for appropriateness, 0.85 for contextualization, and 0.89 for correctness. These scores indicate substantial to almost perfect agreement, demonstrating that our graduate annotators provided reliable and consistent evaluations. For the automatic evaluation, we are actively collecting this result and will update it shortly.

---

> > ### Author Response · Authors · 2025-11-20
> > **Statistical Significance Results.**
> >
> > We apologize for the delay. To depict how our model doesn't just outperform the models on average, we assess whether the improvements of EMPATH over the baselines are statistically meaningful. To this end, we conduct a Wilcoxon signed-rank test across all evaluation metrics for the test set of conversations from respective datasets. Across both CHASE ($500$ conversations) and EmpatheticDialogue ($2.4k$ conversations), EMPATH demonstrates statistically significant improvements over all baseline models on nearly every metric, with Wilcoxon p-values frequently reaching extremely small magnitudes ($10^{-78}–10^{-165}$).
> >
> > $\textbf{CHASE} ($test size$ = 500$ conversations$)$
> > \\begin{array}{llcccccccccc}
> >      \\hline \textbf{Model} & & \textit{PPL}  & \textbf{D-1} & \textbf{D-2}  & \textbf{D-3}  &  \textit{SES}  & \textit{BLEU}  &  \textbf{ROUGE-1 F1-Score}  & \textbf{ROUGE-2 F1-Score}  & \textbf{ROUGE-L F1-Score}  & \textit{METEOR}   \\\\\\hline
> >     GPT-2 & small & 2.54E-78 & 6.44E-68 & 6.05E-36 & 1.97E-64 & 6.32E-84 & 6.32E-84 & 6.32E-84 & 1.64E-78 & 6.32E-84 & 6.32E-84 \\\\\\hline
> >          & medium & 8.28E-79 & 0.1466417 & 5.84E-36 & 3.82E-82 & 6.32E-84 & 2.19E-62 & 6.32E-84 & 6.32E-84 & 6.32E-84 & 6.32E-84 \\\\\\hline
> >          & large & 1.24E-80 & 6.32E-84 & 1.26E-83 & 2.94E-17 & 6.32E-84 & 6.32E-84 & 6.32E-84 & 5.17E-82 & 6.32E-84 & 6.32E-84 \\\\\\hline
> >         DialoGPT & small & 1.69E-80 & 6.32E-84 & 6.32E-84 & 2.43E-81 & 6.32E-84 & 6.32E-84 & 6.32E-84 & 4.32E-82 & 6.32E-84 & 6.32E-84 \\\\\\hline
> >          & medium & 1.00E-47 & 6.32E-84 & 6.32E-84 & 2.94E-17 & 6.32E-84 & 6.32E-84 & 6.32E-84 & 9.59E-83 & 6.32E-84 & 6.32E-84 \\\\\\hline
> >          & large & 4.32E-50 & 6.32E-84 & 6.32E-84 & 1.01E-45 & 6.32E-84 & 6.32E-84 & 6.32E-84 & 1.55E-71 & 6.32E-84 & 6.32E-84 \\\\\\hline
> >         GPT-Neo &  & 5.41E-79 & 6.32E-84 & 6.32E-84 & 2.49E-31 & 6.32E-84 & 6.32E-84 & 6.32E-84 & 6.32E-84 & 6.32E-84 & 6.32E-84 \\\\\\hline
> >         T5-base &  & 8.10E-42 & 6.32E-84 & 5.34E-76 & 7.85E-11 & 6.32E-84 & 6.32E-84 & 6.32E-84 & 6.32E-84 & 6.32E-84 & 6.32E-84 \\\\\\hline
> >         SOTA & EMMA & 8.87E-83 & 6.32E-84 & 3.32E-64 & 0.0003342 & 6.32E-84 & 6.32E-84 & 6.32E-84 & 4.18E-83 & 6.32E-84 & 6.32E-84 \\\\\\hline
> >          & ECTG & 6.32E-84 & 6.32E-84 & 0.0002231 & 1.57E-10 & 6.32E-84 & 6.32E-84 & 6.32E-84 & 1.02E-83 & 6.32E-84 & 6.32E-84 \\\\\\hline
> >          & GATES & 6.32E-84 & 6.32E-84 & 6.32E-84 & 1.07E-65 & 6.32E-84 & 6.32E-84 & 6.32E-84 & 7.80E-84 & 6.32E-84 & 6.32E-84 \\\\\\hline
> >          & GREC & 6.32E-84 & 6.32E-84 & 6.32E-84 & 6.32E-84 & 6.32E-84 & 6.32E-84 & 2.82E-79 & 2.66E-06 & 0.5379567 & 1.36E-70 \\\\\\hline
> >          & SEEC & 7.04E-41 & 6.32E-84 & 4.37E-71 & 4.33E-43 & 6.36E-84 & 4.06E-78 & 7.46E-83 & 2.29E-77 & 6.32E-84 & 3.01E-82 \\\\\\hline
> >         Llama-2 & 7B & 6.32E-84 & 6.32E-84 & 6.32E-84 & 6.32E-84 & 6.32E-84 & 6.32E-84 & 6.32E-84 & 6.32E-84 & 6.32E-84 & 6.32E-84 \\\\\\hline
> >         Llama-3 & 8B & 6.32E-84 & 6.32E-84 & 6.32E-84 & 6.32E-84 & 6.32E-84 & 6.32E-84 & 6.32E-84 & 6.32E-84 & 6.32E-84 & 6.32E-84 \\\\\\hline
> >         Claude & 4.5 & 6.32E-84 & 6.32E-84 & 6.32E-84 & 6.32E-84 & 6.32E-84 & 6.32E-84 & 6.32E-84 & 6.32E-84 & 6.32E-84 & 6.32E-84 \\\\\\hline
> >         Gemini & 2.5 & 6.32E-84 & 6.32E-84 & 6.32E-84 & 6.32E-84 & 6.32E-84 & 6.32E-84 & 6.32E-84 & 6.32E-84 & 6.32E-84 & 6.32E-84 \\\\\\hline
> >     \end{array}

---

> > > ### Author Response · Authors · 2025-11-20
> > > **Statistical Significance Results (continued).**
> > >
> > > $\textbf{EmpatheticDialogue} ($test size$ = 2.4$k conversations$)$
> > > \\begin{array}{llcccccccccc}
> > >      \\hline \textbf{Model} & & \textit{PPL}  & \textbf{D-1} & \textbf{D-2}  & \textbf{D-3}  &  \textit{SES}  & \textit{BLEU}  &  \textbf{ROUGE-1 F1-Score}  & \textbf{ROUGE-2 F1-Score}  & \textbf{ROUGE-L F1-Score}  & \textit{METEOR}   \\\\\\hline
> > > 	GPT-2 & small & 2.55E-155 & 4.87E-159 & 5.17E-41 & 1.97E-163 & 1.66E-165 & 1.66E-165 & 1.66E-165 & 1.66E-165 & 1.66E-165 & 1.66E-165 \\\\\\hline
> > >         &	medium & 1.66E-165 & 1.66E-165 & 1.06E-164 & 0.0020143 & 1.66E-165 & 1.66E-165 & 1.66E-165 & 1.66E-165 & 1.66E-165 & 1.66E-165\\\\\\hline
> > >         &	large & 1.66E-165 & 1.66E-165 & 1.66E-165 & 2.52E-165 & 1.66E-165 & 1.66E-165 & 1.66E-165 & 1.66E-165 & 1.66E-165 & 1.66E-165\\\\\\hline
> > >         DialoGPT & small & 1.89E-125 & 1.66E-165 & 3.25E-102 & 0.1787781 & 1.66E-165 & 1.66E-165 & 1.66E-165 & 1.66E-165 & 1.66E-165 & 1.66E-165\\\\\\hline
> > >         &	medium & 4.80E-143 & 1.66E-165 & 2.80E-163 & 9.97E-69 & 1.66E-165 & 1.66E-165 & 1.66E-165 & 1.66E-165 & 1.66E-165 & 1.66E-165\\\\\\hline
> > >         &	large & 1.66E-165 & 2.69E-145 & 4.19E-27 & 5.54E-162 & 1.66E-165 & 1.66E-165 & 1.66E-165 & 1.66E-165 & 1.66E-165 & 1.66E-165\\\\\\hline
> > >         GPT-Neo & & 1.66E-165 & 1.66E-165 & 1.66E-165 & 1.66E-165 & 1.66E-165 & 1.66E-165 & 1.66E-165 & 1.66E-165 & 1.66E-165 & 1.66E-165\\\\\\hline
> > >         T5-base & & 1.48E-31 & 1.66E-165 & 1.66E-165 & 3.23E-164 & 1.66E-165 & 1.66E-165 & 1.66E-165 & 1.66E-165 & 1.66E-165 & 1.66E-165\\\\\\hline
> > >         SOTA & EMMA & 1.66E-165 & 1.66E-165 & 1.66E-165 & 6.55E-165 & 1.66E-165 & 1.66E-165 & 1.66E-165 & 1.66E-165 & 1.66E-165 & 1.66E-165\\\\\\hline
> > >         &	ECTG & 1.66E-165 & 1.66E-165 & 1.66E-165 & 9.38E-134 & 1.66E-165 & 1.66E-165 & 1.66E-165 & 1.66E-165 & 1.66E-165 & 1.66E-165\\\\\\hline
> > >         &	GATES & 1.66E-165 & 1.66E-165 & 2.17E-144 & 4.76E-79 & 1.66E-165 & 3.33E-139 & 1.66E-165 & 1.66E-165 & 1.66E-165 & 1.66E-165\\\\\\hline
> > >         &	GREC & 1.66E-165 & 1.66E-165 & 1.66E-165 & 2.50E-165 & 4.99E-165 & 2.54E-165 & 9.59E-164 & 1.66E-165 & 3.09E-162 & 0.0014499\\\\\\hline
> > >         &	SEEC & 0.3810731 & 1.66E-165 & 6.37E-53 & 2.61E-14 & 7.63E-55 & 2.54E-165 & 3.21E-164 & 2.75E-157 & 1.66E-165 & 1.66E-165\\\\\\hline
> > >         Llama-2 & 7B & 1.66E-165 & 1.66E-165 & 1.66E-165 & 1.66E-165 & 1.66E-165 & 1.66E-165 & 1.66E-165 & 1.66E-165 & 1.66E-165 & 1.66E-165\\\\\\hline
> > >         Llama-3 & 8B & 1.66E-165 & 1.66E-165 & 1.66E-165 & 1.66E-165 & 1.66E-165 & 1.66E-165 & 1.66E-165 & 1.66E-165 & 1.66E-165 & 1.66E-165\\\\\\hline
> > >         Claude & 4.5 & 1.66E-165 & 1.66E-165 & 1.66E-165 & 1.66E-165 & 1.71E-165 & 1.66E-165 & 1.66E-165 & 1.66E-165 & 1.66E-165 & 1.66E-165\\\\\\hline
> > >         Gemini & 2.5 & 1.66E-165 & 1.66E-165 & 1.66E-165 & 1.66E-165 & 1.66E-165 & 1.66E-165 & 1.66E-165 & 1.66E-165 & 1.66E-165 & 1.66E-165\\\\\\hline
> > > \end{array}

---

### Official Review · Reviewer_E3kz · 2025-10-28

**Soundness:** 2
**Presentation:** 2
**Contribution:** 3
**Rating:** 4
**Confidence:** 4

**Summary:**

This paper proposes a full pipeline for endowing dialogue systems with affective (not only cognitive) empathy. The authors (i) construct a large “empathic statements” corpus by sampling ChatGPT across ~500 rounds (upper bound ≈125k sentences; after filtering ~111k unique) and then applying two-stage paraphrasing with PEGASUS-Large to reach ~81 M sentence, with Perspective-API filtering and 1% human-in-the-loop active learning; (ii) continue pre-training a T5-base model on this corpus to obtain E-T5; and (iii) introduce EMPATH, a decoding/learning framework that combines multi-emotion detection, explanation-based emotion-cause extraction via Integrated Gradients, and a triple loss (emotion-distribution KL, contextual cosine similarity, and an empathy-classifier MSE) within an iterative generation loop.

Evaluation: Empathetic response generation (ERG) is reported on two datasets (EmpatheticDialog, CHASE). Emotion-cause extraction (ECE) is evaluated on RECCON. EMPATH outperforms diverse baselines on automatic metrics (e.g., perplexity, BLEU, Distinct-n, SES) and human preference ratings. Ablations indicate all modules contribute.

**Strengths:**

1. Resource contribution. The paper releases (and documents filtering for) an 80M-scale corpus explicitly targeting affective empathy, which is rare and potentially valuable to the community.

2. Methodological novelty. Coupling multi-emotion awareness with explanation-based cause extraction and a unified triple loss is conceptually clean and addresses the “template empathy” failure mode seen in prior work.

3. Empirical coverage for tasks. ERG and ECE are both covered, with ablations that clarify the effect of affective pre-training (E-T5), multi-emotion gating, explanation-based causes, and the loss design.

**Weaknesses:**

1. Baseline fairness. E-T5 enjoys a substantial extra pre-training advantage (~8.5 GB) whereas several baselines are only fine-tuned. There is no comparison to size-matched strong baselines (e.g., FLAN-T5-base) or to frontier LLMs run at high temperature for diversity.

2. Diversity claim under-evidenced. The stated motivation that PEGASUS paraphrasing yields more diverse outputs than ChatGPT is not empirically verified via Distinct-n, self-BLEU, or MAUVE comparisons.

3. Human evaluation scale. Only six graduate annotators are used, and inter-rater reliability is not reported, limiting the strength of human-study conclusions.

**Questions:**

1. Please clarify the fairness of the baseline comparisons.

2. Please specify the exact model version and hyperparameters used for ChatGPT, Claude, and Gemini (precise model versions,  decoding parameters such as temperature/top-p/max tokens, context length, prompts, and seeds).

---

> ### Author Response · Authors · 2025-11-18
> **Response to reviewer**
>
> Thank you so much for your positive response to our work. Based on your pointed out weaknesses, we have the following justification and amendments. We apologize for the delay caused due to conducting a few experiments.
> 1. $\\textbf{Human evaluation details:}$ We ensured annotator quality by administering the McLaren[1] screening test and selecting only those who scored above 75%, ensuring a consistent understanding of empathy-related concepts. In addition, we report inter-rater reliability using Fleiss’ Kappa, obtaining values of 0.88 for empathy, 0.82 for appropriateness, 0.85 for contextualization, and 0.89 for correctness. These scores indicate substantial to almost perfect agreement, demonstrating that our graduate annotators provided reliable and consistent evaluations.
> 2. $\\textbf{Model Details:}$ We have updated the manuscript to specify the exact model versions and decoding parameters used. For generation, we employ ChatGPT-3.5, and for comparison, we use Claude 4.5 and Gemini 2.5. We also include all decoding and training hyperparameters: "epochs$=10$ , batch size$=16$, the Adam optimizer, learning rate$=1e-3$, $top\_k=50$, $top\_p=0.95$, temperature$=0.85$, $max tokens=200$, context length$=$ default per API and a uniform prompt across LLMs: ``Generate an empathetic (not sympathetic) response to the following conversation") to ensure comparability."
> 3. $\\textbf{Diversity Claim:}$ We have now empirically validated our diversity claim by including Distinct-n and self-BLEU comparisons between PEGASUS and several popular LLMs used for paraphrasing. The results, added to the Appendix, show that PEGASUS consistently achieves higher Distinct-n scores and competitive self-BLEU, supporting our choice of PEGASUS for generating more diverse paraphrases.
> \\begin{array}{|c|c|c|c|c|}
> \\hline \textbf{Model} & \textbf{Distinct-1} & \textbf{Distinct-2} & \textbf{Distinct-3} & \textbf{Self-BLEU} \\\\\\hline
> Pegasus & \textbf{0.588} & \textbf{0.926} & \textbf{0.929} & \textbf{0.851} \\\\\\hline
> FLAN-T5 & 0.504 & 0.888 & 0.910 & 0.839 \\\\\\hline
> GPT-4 & 0.529 & 0.886 & 0.917 & 0.767 \\\\\\hline
> Gemini 2.5 & 0.496 & 0.904 & 0.922 & 0.423 \\\\\\hline
> Claude 4.5 & 0.492 & 0.840 & 0.868 & 0.716 \\\\\\hline
> \\end{array}
> 4. $\\textbf{Decoding Fairness:}$ Our models were evaluated at a relatively high temperature (0.85). We additionally experimented with higher temperatures (0.9), but observed no improvements in affective-empathy metrics. We also note that frontier LLMs perform comparably to Claude, Gemini, and Llama-3 in this setting, and we have added a clarification in the paper to avoid ambiguity.
>
>     $\textbf{Baseline Fairness:}$ Our goal is to isolate the impact of incorporating affective-empathy signals, and therefore we compare our E-T5 models against the raw versions of standard baselines to demonstrate how effectively the additional pre-training data enhances empathetic response quality.
>
>    In response to the reviewer’s suggestion, we have now added comparisons where a few more baselines are further pre-trained on our dataset (Table 1 in the manuscript), providing more assessment of the importance of this dataset. Importantly, our results demonstrate that the 8.5GB affective-empathy corpus is already large enough to induce $\textit{substantial and measurable improvements}$ in lower-parameter models such as GPT-2 and DialoGPT (small and medium). However, extending this pre-training to larger models would require proportionally larger datasets—something our data-generation pipeline can readily support, but which becomes computationally expensive at scale. We now emphasize this limitation and its implications in the manuscript. Also, a comparison with prompting has now been added in the Appendix.
>
> [1] K. McLaren. The Art of Empathy: A Complete Guide to Life’s Most Essential Skill. Sounds True, 2013.
> ISBN 9781622030675.

---

> ### Author Response · Authors · 2025-11-18
> **Results of Additional Pre-training**
>
> \\begin{array}{lcc|cc|cc|cc|cc|cc|cc|cc|cc|cc|cc}
> \\hline \textbf{Model} & \textit{PPL} (w/o) & \textit{PPL} (w) & \textbf{D-1} (w/o) & \textit{D-1} (w) & \textbf{D-2} (w/o) & \textit{D-2} (w) & \textbf{D-3} (w/o) & \textit{D-3} (w) &  \textit{SES} (w/o) & \textit{SES} (w) & \textit{BLEU} (w/o) & \textit{BLEU} (w) &  \textit{AL} (w/o) & \textit{AL} (w) & \textbf{ROUGE-1 F1-Score} (w/o) & \textit{ROUGE-1 F1-Score} (w) & \textbf{ROUGE-2 F1-Score} (w/o) & \textit{ROUGE-2 F1-Score} (w) & \textbf{ROUGE-L F1-Score} (w/o) & \textit{ROUGE-L F1-Score} (w) & \textit{METEOR} (w/o) & \textit{METEOR} (w)
> \\\\\\hline
>     GPT-2 (small) & 18.251 & 15.778 & 0.698 & 0.708 & 0.926 & 0.933 & 0.896 & 0.924 & 0.67 & 0.782 & 0.250 & 0.250 & 4 & 13 & 0.261 & 0.349 & 0.142 & 0.349 & 0.25 & 0.349 & 0.146 & 0.156  \\\\\\hline
>     GPT-2 (medium) & 18.539 & 15.247 & 0.797 & 0.799 & 0.928 & 0.934 & 0.856 & 0.916 & 0.685 & 0.793 & 0.333 & 0.333 & 4 & 17 & 0.106 & 0.311 & 0.008 & 0.347 & 0.117 & 0.333 & 0.043 & 0.142 \\\\\\hline
>     DialoGPT (small) & 18.775 & 14.975 & 0.431 & 0.646 & 0.768 & 0.821 & 0.847 & 0.911 & 0.555 & 0.689 & 0.095 & 0.115 & 14 & 16 & 0.129 & 0.187 & 0.105 & 0.187 & 0.103 & 0.187 & 0.095 & 0.182 \\\\\\hline
>     DialoGPT (medium) & 16.334 & 14.555 & 0.352 & 0.583 & 0.792 & 0.803 & 0.929 & 0.931 & 0.239 & 0.444 & 0.058 & 0.095 & 13 & 16 & 0.280 & 0.261 & 0.114 & 0.142 & 0.212 & 0.250 & 0.260 & 0.339  \\\\\\hline
> \\end{array}
>
> \\begin{array}{lcc|cc|cc|cc|cc|cc|cc|cc|cc|cc|cc}
> \\hline \textbf{Model} & \textit{PPL} (w/o) & \textit{PPL} (w) & \textbf{D-1} (w/o) & \textit{D-1} (w) & \textbf{D-2} (w/o) & \textit{D-2} (w) & \textbf{D-3} (w/o) & \textit{D-3} (w) &  \textit{SES} (w/o) & \textit{SES} (w) & \textit{BLEU} (w/o) & \textit{BLEU} (w) &  \textit{AL} (w/o) & \textit{AL} (w) & \textbf{ROUGE-1 F1-Score} (w/o) & \textit{ROUGE-1 F1-Score} (w) & \textbf{ROUGE-2 F1-Score} (w/o) & \textit{ROUGE-2 F1-Score} (w) & \textbf{ROUGE-L F1-Score} (w/o) & \textit{ROUGE-L F1-Score} (w) & \textit{METEOR} (w/o) & \textit{METEOR} (w)
> \\\\\\hline
>     GPT-2 (small)  & 17.342 & 14.365 & 0.705 & 0.741 & 0.941 & 0.949 & 0.882 & 0.892 & 0.065 & 0.413 & 0.180 & 0.321 & 6 & 15 & 0.066 & 0.146 & 0.171 & 0.111 & 0.066 & 0.2 & 0.02 & 0.151\\\\\\hline
>     GPT-2 (medium)  & 22.483 & 18.952 & 0.328 & 0.597 & 0.863 & 0.902 & 0.958 & 0.960 & 0.696 & 0.753 & 0.142 & 1.347 & 11 & 16 & 0.076 & 0.129 & 0.166 & 0.205 & 0.076 & 0.103 & 0.094 & 0.172\\\\\\hline
>     % \citep{10128674}\\\\\\hline
>     DialoGPT (small)  & 15.037 & 12.983 & 0.440 & 0.567 & 0.920 & 0.935 & 0.960 & 0.961 & 0.320 & 0.549 & 0.209 & 0.333 & 11 & 17 & 0.162 & 0.166 & 0.090 & 0.166 & 0.162 & 0.166 & 0.080 & 0.157\\\\\\hline
>     DialoGPT (medium) & 16.113 & 12.746 & 0.416 & 0.582 & 0.875 & 0.928 & 0.937 & 0.955 & 0.551 & 0.633 & 0.245 & 0.329 & 10 & 18 & 0.232 & 0.286 & 0.098 & 0.280 & 0.143 & 0.286 & 0.031 & 0.120 \\\\\\hline
> \\end{array}

---

> > ### Comment · Reviewer_E3kz · 2025-11-26
> >
> > Thank you for the detailed and constructive response, as well as for running additional experiments.
> >
> > Several of my earlier concerns are now partially alleviated. In particular, the additional information on annotator screening and the reported Fleiss’ Kappa scores address my worry about the reliability of the human evaluation. The clarification of the exact LLM versions and decoding hyperparameters, as well as the new Distinct-n / self-BLEU analysis comparing PEGASUS to popular LLM paraphrasers, also strengthen the empirical grounding of the claims about diversity. I appreciate, too, the extra experiments where other baselines are further pre-trained on the proposed dataset, which give a clearer picture of how much the affective-empathy corpus itself contributes.
> >
> > That said, some limitations around baseline fairness and scale remain: E-T5 continues to benefit from substantial additional pre-training compared to several of the main baselines, and fully size- and data-matched comparisons to strong instruction-tuned models are still somewhat limited. Overall, the rebuttal improves my understanding of the work and increases my confidence that the main claims are supported.

---

> > > ### Author Response · Authors · 2025-11-26
> > > **Further Clarifications**
> > >
> > > Thank you again for your thoughtful follow-up. We are glad that the clarifications regarding diversity analysis, human-evaluation reliability, hyperparameter transparency, and additional baseline pre-training addressed your earlier concerns.
> > >
> > > $\textbf{Regarding the baseline fairness point}$: Our primary goal is not to show that E-T5 outperforms baseline models per se, but rather to demonstrate that incorporating affective-empathy during pre-training consistently enhances empathy-related behavior, even on non-empathetic dialogue datasets (Table $4$). We further support this by showing how this dataset improves various existing cause-related tasks and models of different sizes (Table $1$).
> > >
> > > Additionally, in Tables $4-6$, we compare E-T5 with models in their raw form (i.e., without additional pre-training on our dataset). Low-parameter models (GPT-small/medium, DialoGPT-small/medium) tend to exhibit empathy even in inappropriate contexts, while higher-parameter models show little to no change in behavior, as noted in our limitations section and Appendix G, more data would be required to affect these models. So, in these comparisons we primarily illustrate how affective empathy actually improves the overall empathy of the response.
> > >
> > > We appreciate your constructive engagement and hope this contextualization addresses your remaining concern.

---

### Official Review · Reviewer_vqtz · 2025-10-30

**Soundness:** 2
**Presentation:** 2
**Contribution:** 3
**Rating:** 4
**Confidence:** 4

**Summary:**

The paper proposes a novel framework, EMPATH, to enhance empathetic response generation (ERG) in dialogue systems. It addresses limitations in existing models that exhibit only cognitive empathy and produce generic, unrelatable responses. The authors construct an 8.5GB affective empathy dataset using ChatGPT and Pegasus-based paraphrasing to pre-train a T5-based model. EMPATH combines affective and cognitive empathy, uses explainability for multi-emotion cause extraction, and introduces an emotion-aware contextual loss. Extensive experiments show superior empathetic, contextually relevant, and diverse responses across benchmark datasets.

**Strengths:**

- The paper correctly identifies an important and underexplored limitation in current ERG systems—the absence of affective empathy in responses.
- The authors perform extensive evaluation using both quantitative metrics (BLEU, ROUGE, Distinct-n, etc.) and human assessments, providing evidence of performance gains.

**Weaknesses:**

- Writing:
  - The paper is difficult to follow due to the mixed presentation of methodology and implementation details.
  - Many design choices in dataset construction and model formulation are not well motivated. Readers are often left uncertain why a particular step was necessary or how it contributes to affective empathy.
  - Some sentences are very confusing, e.g., “we additionally pre-train the T5 model due to the open-source unavailability of LLMs and its strong performance.” Overall writing quality significantly hinders comprehension.
- Evaluation: Although the paper poses the lack of affective empathy as a key research question, the evaluation does not directly measure it. The presented metrics (BLEU, Distinct-n, etc.) and corpus-level analysis do not show whether responses are indeed more affective rather than merely cognitive. Human evaluation focusing on empathy type or emotional depth would strengthen the claims.
- The core claim—that affective empathy can be explicitly modelled is not contrasted against simpler or more intuitive alternatives. For instance, prompting LLMs with explicit instructions to generate affective (and avoid purely cognitive) responses could serve as an important baseline. Without such comparisons, it is unclear whether EMPATH’s complexity is necessary.

**Questions:**

See the above comments.

---

> ### Author Response · Authors · 2025-11-18
>
> Thank you so much for your positive response to our work. Based on your pointed out weaknesses, we have the following justification and amendments. We apologize for the delay caused due to conducting a few experiments.
> 1. $\textbf{Affective Empathy evaluation}$: We have expanded our human evaluation to explicitly measure the type of empathy expressed in model responses. Annotators now categorize each response as $\textit{Affective}$, $\textit{Cognitive}$, $\textit{Generic}$, or $\textit{IDK}$, enabling a direct assessment of affective empathy. The results of this new evaluation have been incorporated into Table 6 in the manuscript.
> \\begin{array}{|c|c|c|c|c|c|}
> \\hline \textbf{Model} & & \textbf{Affective Empathy} & \textbf{Cognitive Empathy} & \textbf{Generic} & \textbf{IDK} \\\\\\hline
> GPT-2  & small  & 12 & 37 & 48 & 3 \\\\\\hline
>  &  medium  & 7 & 33 & 54 & 6\\\\\\hline
>  &  large  & 5 & 41 & 47 & 7\\\\\\hline
> DialoGPT  & small  & 7 & 28 & 45 & 20\\\\\\hline
>  & medium  & 10 & 24 & 53 & 13\\\\\\hline
>   & large  & 3 & 31 & 50 & 16\\\\\\hline
>  GPT-Neo  &  & 8 & 17 & 73 & 2\\\\\\hline
>  T5-base  &  & 20 & 29 & 26 & 25\\\\\\hline
>  E-T5  & medium  & 62 & 18 & 15 & 5\\\\\\hline
>   & large  & 74 & 8 & 15 & 3\\\\\\hline
> SOTA  & EMMA  & 12 & 25 & 53 & 10\\\\\\hline
>   & ECTG  & 14 & 34 & 35 & 17\\\\\\hline
>   & GATES  & 9 & 38 & 48 & 5\\\\\\hline
>   & GREC  & 17 & 27 & 45 & 11\\\\\\hline
>   & SEEC  & 29 & 48 & 5 & 18\\\\\\hline
>   Llama-2  & 7B  & 20 & 52 & 26 & 2\\\\\\hline
>  Llama-3  & 8B  & 23 & 58 & 17 & 2\\\\\\hline
>  Claude  & 4.5  & 14 & 60 & 21 & 5\\\\\\hline
>  Gemini  & 2.5  & 27 & 63 & 9 & 1\\\\\\hline
>  EMPATH  & medium  & 63 & 18 & 15 & 4\\\\\\hline
>  & large  & 75 & 11 & 11 & 3\\\\\\hline
> \\end{array}
> 2. $\textbf{Dataset details}$: Because of space constraints, the dataset section was originally presented in a condensed form. We agree that additional motivation would improve clarity, so we have revised the main text to better justify key design choices. We also include a dedicated Appendix that provides step-by-step explanations of each component, why it was chosen, and its contribution to affective-empathy modeling.
> 3. $\textbf{Re-organisation of implementation details}$: We appreciate this observation and carefully considered restructuring the sections. However, we found that separating methodology and implementation details disrupted the logical flow of how E-T5 is introduced and developed. Presenting them together allows the reader to follow the progression from conceptual motivation to practical realization more coherently.
> 4. $\textbf{Prompting LLMs}$: We apologize for this confusion. We did include prompting: all LLMs are evaluated using an explicit instruction: "Generate an empathetic (not sympathetic) response to the following conversation", to encourage affective rather than purely cognitive empathy. We have now clarified this in the experimental setup section to make the comparison explicit and to show that EMPATH’s improvements go beyond what can be achieved through prompting alone. We also note that combining such prompting strategies with our synthetic dataset may further enhance affective-empathy performance, and we highlight this as a promising direction for future work.

---

### Author Response · Authors · 2025-12-01
**Summary of rebuttal based on Reviewer vqtz**

We thank everyone for overseeing our submission and provide below a concise summary of the changes implemented in response to the reviewers’ valuable feedback.

1. $\\textcolor{blue}{\text{The details provided regarding dataset construction, formulation, and the rationale behind various choices were insufficient.}}$

$\\textbf{Author’s Response}$: We appreciate this feedback. In response, we have added more comprehensive details on the dataset construction process to ensure reproducibility. Specifically, we now explain the rationale behind choosing particular models. For example, we selected ChatGPT for initial generation based on the findings of Wu et al., and we clarify the choice of the Pegasus model with an accompanying experiment. Additionally, we have included a flow diagram to visually represent the contribution of each model at every step of the process.

$\\textcolor{green}{\text{Section Changed}}$: $\\textbf{Section 3 and Appendix D}$

2. $\\textcolor{blue}{\text{The reviewer suggested that a human evaluation focusing on empathy type or emotional depth would strengthen the claims.}}$

$\\textbf{Author’s Response}$: This valuable suggestion has now been incorporated as a part of the human evaluation, where the response is either affective, cognitive, generic or the verifier does not know. These improved results help back our claim that EMPATH produces responses that are indeed more affective rather than merely cognitive.

$\\textcolor{green}{\text{Section Changed}}$: $\\textbf{Table 6 on Page 9 and respective descriptions have been improved in Appendix B.}$

3. $\\textcolor{blue}{\text{They suggested prompting LLMs with explicit instructions to generate affective (and avoid purely cognitive) responses to serve as an important baseline.}}$

$\\textbf{Author’s Response}$: We have followed this suggestion and clarified that the results presented in Tables 4–6 include experiments where LLMs are prompted in a zero-shot scenario. Furthermore, we have extended the analysis to include one-shot and five-shot (few-shot) prompt configurations, as well as explored the effect of temperature on the output of frontier LLMs. These modifications strengthen our study by providing a clear baseline and demonstrating the superior performance of our model.

$\\textcolor{green}{\text{Section Changed}}$: $\\textbf{Appendix G}$

4. $\\textcolor{blue}{\text{The reviewer raised concerns about the overall writing quality of the manuscript.}}$

$\\textbf{Author’s Response}$: In response to this comment, we have thoroughly revised the manuscript for clarity and improved the writing, including the specific sentence identified by the reviewer.

$\\textcolor{green}{\text{Section Changed}}$: $\\textbf{Line 190 (as indicated by the reviewer)}$

$\\textcolor{green}{\text{To summarize, we have addressed all of the reviewers’ concerns and incorporated their valuable suggestions into the revised manuscript.}}$

---

> ### Author Response · Authors · 2025-12-01
> **Summary of rebuttal based on Reviewer E3kz**
>
> 1. $\\textcolor{blue}{\text{The PEGASUS paraphrasing yields more diverse outputs than ChatGPT is under-evidenced.}}$
>
> $\\textbf{Author’s Response}$: We appreciate this feedback. In response, we have now added a clarifying explanation and an accompanying experiment for the choice of the Pegasus model that demonstrates how it generates more diverse statements after paraphrasing than frontier paraphrasing models.
>
> $\\textcolor{green}{\text{Section Changed}}$: $\\textbf{Appendix D}$
>
> 2. $\\textcolor{blue}{\text{The reviewer felt that human-study conclusions were weak due to the presence of only six graduate annotators and that inter-rater reliability is not reported.}}$
>
> $\\textbf{Author’s Response}$: We clarified that the annotator quality was ensured by administering the McLaren screening test and selecting only those who scored above 75\%, ensuring a consistent understanding of empathy-related concepts. In addition, we report inter-rater reliability using Fleiss’ Kappa, obtaining values of 0.88 for empathy, 0.82 for appropriateness, 0.85 for contextualization, and 0.89 for correctness. These scores indicate substantial to almost perfect agreement, demonstrating that our graduate annotators provided reliable and consistent evaluations.
>
> $\\textcolor{green}{\text{Section Changed}}$: $\\textbf{Table 6 on Page 9 and Section 5.3.}$
>
> 3. $\\textcolor{blue}{\text{They suggested specifying the exact model version and hyperparameters used.}}$
>
> $\\textbf{Author’s Response}$: Following the reviewer’s suggestion, we added full details on the model versions and hyperparameters used across all language models. The revisions explicitly list temperature, top-p, maximum tokens, context length, and prompts utilized in the experiments.
>
> $\\textcolor{green}{\text{Section Changed}}$: $\\textbf{Section 5.1 and Appendix G}$
>
> 4. $\\textcolor{blue}{\text{The reviewer needed clarity on the fairness of the baseline comparisons.}}$
>
> $\\textbf{Author’s Response}$: Our primary goal is not to show that E-T5 outperforms baseline models per se, but rather to demonstrate that incorporating affective-empathy during pre-training consistently enhances empathy-related behavior, even on non-empathetic dialogue datasets (Table 4). We further support this by showing how this dataset improves various existing cause-related tasks and models of different sizes (Table 1).
> Additionally, in Tables 4-6, we compare E-T5 with models in their raw form (i.e., without additional pre-training on our dataset). Low-parameter models (GPT-small/medium, DialoGPT-small/medium) tend to exhibit empathy even in inappropriate contexts, while higher-parameter models show little to no change in behavior, as noted in our limitations section and Appendix G, more data would be required to affect these models. So, in these comparisons we primarily illustrate how affective empathy actually improves the overall empathy of the response.
>
> $\\textcolor{green}{\text{Section Changed}}$: $\\textbf{Appendix G and Limitation}$
>
> $\\textcolor{green}{\text{To summarize, we were able to clear most of the reviewers’ concerns (as acknowledged by them) and incorporated their valuable suggestions into the revised manuscript.}}$

---

> ### Author Response · Authors · 2025-12-01
> **Summary of rebuttal based on Reviewer NJTA**
>
> 1. $\\textcolor{blue}{\text{The reviewer found the statistical significance analysis of the results missing.}}$
>
> $\\textbf{Author’s Response}$: This valuable suggestion has been incorporated in our work to depict how our model consistently outperforms all the baselines using the Wilcoxon signed-rank test across all evaluation metrics for the test set of conversations from both empathetic dialogue datasets (CHASE and EmpatheticDialogue).
>
> $\\textcolor{green}{\text{Section Changed}}$: $\\textbf{Appendix F}$
>
> 2. $\\textcolor{blue}{\text{The reviewer pointed out that the process for tuning hyperparameters (λ₁/λ₂/λ₃) in the emotion-aware contextual loss is insufficiently described.}}$
>
> $\\textbf{Author’s Response}$: This point has been keenly noted and clarified in the manuscript now where we now mention that during optimization, they are updated jointly with the model parameters using the Adam optimizer, based on the gradients of the total loss with respect to each λ.
>
> $\\textcolor{green}{\text{Section Changed}}$: $\\textbf{Line 265 on Page 6.}$
>
> 3. $\\textcolor{blue}{\text{They suggested that there is no evaluation of whether EMPATH can avoid expressing empathy when it is inappropriate, which may lead to misalignment with user intent.}}$
>
> $\\textbf{Author’s Response}$: To address the reviewer’s concern about ensuring that EMPATH does not express empathy in inappropriate contexts, we added several illustrative examples in the appendix. These cases show that EMPATH preserves this important behavioral constraint. We further explain that T5 was selected as the base model because its parameter scale helps retain core conversational behaviors, including the suppression of empathy when inappropriate. In contrast, lower-parameter models fine-tuned on the same dataset tend to overgeneralize and produce empathetic responses regardless of context.
>
> $\\textcolor{green}{\text{Section Changed}}$: $\\textbf{Appendix G}$
>
> 4. $\\textcolor{blue}{\text{The reviewer raised concerns that the study does not consider multilingual or cross-cultural dimensions of empathy.}}$
>
> $\\textbf{Author’s Response}$: We clarified that currently, our work focuses exclusively on English. We agree that exploring multilingual and cross-cultural dimensions of empathy is an important future direction. In principle, large-scale empathetic data in other languages could be generated using LLMs like ChatGPT, as we do for English. However, current paraphrasing tools for other languages are limited, making large-scale dataset creation challenging at this stage.
>
> 5. $\\textcolor{blue}{\text{The reviewer raised concerns that the real-world validity of the synthetic dataset remains untested.}}$
>
> $\\textbf{Author’s Response}$: Regarding this concern, we would like to clarify that we have addressed it in Tables 4 and 5, where, after pre-training on affective empathy, the model is fine-tuned on datasets containing real user conversations. Across all such datasets, our model consistently outperforms the baselines. Human evaluation also supports these findings, indicating that the synthetic pre-training transfers effectively to real-world dialogues.
>
> $\\textcolor{green}{\text{To summarize, we have addressed all of the reviewers’ concerns and incorporated their valuable suggestions into the revised manuscript.}}$

---

### Meta-Review · Area_Chair_HYdi · 2026-01-07

**Summary:**

The paper tackles key challenges in Empathetic Response Generation (ERG), including limited affective empathy, generic responses, and low user relatability. To address these, the authors propose EMPATH, which enhances affective empathy through additional pre-training on a newly curated dataset. EMPATH also reduces generic responses via a novel loss function that leverages both conversation history and target responses, and improves user relatability by modeling multiple emotions and their underlying causes with explainability. Experiments demonstrate that EMPATH outperforms existing methods on benchmark ERG datasets across multiple metrics.

While the proposed benchmark is interesting, some metrics, such as PPL, raise concerns, and the methodology behind them is not clearly explained. Additionally, the new results reported in the rebuttal lack sufficient explanation. The technical novelty is also a critical issue on this new proposed benchmark. Due to these issues, the evaluation remains unconvincing, and I recommend rejection.

**Reviewer Concerns:**

The fairness of the evaluation is still not clearly explained. Some of the details are not provided in the paper, which makes the work not reproducible.

**Reviewer Scores:**

Reviewer E3kz may reconsider the score.

---

### Decision · Program_Chairs · 2026-01-26

Reject